# Recent Advances in the Nutritional Value, Chemical Compositions, Pharmacological Activity, and Application Value of *Orychophragmus violaceus*: A Comprehensive Review

**DOI:** 10.3390/molecules29061314

**Published:** 2024-03-15

**Authors:** Xiaolan Chen, Guangjie Zhang, Wenjin Cui, Chunbo Ge, Bin Li, Min Li, Shuchen Liu, Lin Wang

**Affiliations:** 1School of Pharmacy, Guangdong Pharmaceutical University, Guangzhou 510006, China; nancycxl@163.com; 2Beijing Institute of Radiation Medicine, Beijing 100850, China; zhanggj410@sina.com (G.Z.); cuiwj112233@163.com (W.C.); gechunb@163.com (C.G.); jkylibin@hotmail.com (B.L.); limin82057@163.com (M.L.)

**Keywords:** *Orychophragmus violaceus*, Brassicaceae, chemical compositions, alkaloids, ferroptosis, pharmacological activity

## Abstract

*Orychophragmus violaceus* (L.) O. E. Schulz (Brassicaceae) is widely distributed and plentiful in China and has been widely used for its application in ornamental, oil, ecology, foraging, and food. Recent studies have revealed that the main components of *Orychophragmus violaceus* include flavonoids, alkaloids, phenylpropanoids, phenolic acids, terpenoids, etc., which have pharmacological activities such as antioxidation, antiradiation, antitumor, hepatic protection, antiferroptosis, anti-inflammatory, and antibacterial. In this paper, the nutritional value, chemical compositions, pharmacological activity, and application value of *Orychophragmus violaceus* are summarized by referring to the relevant domestic and international literature to provide a reference for further research, development, and utilization of *Orychophragmus violaceus* in the future.

## 1. Introduction

*Orychophragmus violaceus*, belonging to the *Orychophragmus Genus*, is a plant that falls within the Brassicaceae family. It is recognized by various common names such as “Er-yue-lan” and “Zi-jin-cao”. This species is native to China and can be found as both an annual and perennial plant. Its distribution is widespread, with occurrences scattered throughout the southwest and northeast regions of China. It mostly grows in plains, grasslands, mountain slopes, roadsides, or woodland edges [1]. Legend has it that Zhuge Liang picked a roadside plant to feed his troops on their way to war because of a lack of grain. Chen Zhenhui et al. [2] suggested that the *Orychophragmus Genus* has only one species, namely *O. violaceus*, while subspecies include *O. hupehensis+ O. taibaiensis*, and *O. diffusus*.

The predominant height of *Orychophragmus violaceus* plants ranges from 30 to 50 cm, exhibiting erect stems and a singular central stem configuration. Flowering from April to May, the initial flowers are mostly blue-purple or light red and eventually turn white in lax racemes. The pollen polar views of *O. violaceus* are trilobed round and the pollen equatorial views are prolate, which have three colpi. The longitudinal growth of the pod is faster than the transverse broadening and the pod length is about 28 days after anthesis stereotypes. The fruiting periods are May–June, the fruit is long, silique cylindrical, and the seeds are ovate and rounded. *O. violaceus* exhibits high cold resistance and shade tolerance, remaining evergreen even during the winter months [3,4,5,6]. *O. violaceus* can be found everywhere along the roadsides in China and is commonly used as an ornamental plant to both prevent soil erosion and beautify the environment.

It was first recorded in “Hou Han Shu: Ji: Xiao Huan Di Ji”: “It’s warm-natured, pungent, bitter and sweet in taste, and is effective in treating jaundice in children, deep-rooted breast carbuncles, and common fever and other diseases”.

The tender leaves and stems of *O. violaceus* are edible, containing various nutrients such as carotene, protein, and vitamins. The whole herb of *O. violaceus* is employed as a medicinal resource, known for its ability to harmonize the intestine-stomach, promote diuresis, and facilitate detoxification [7]. Furthermore, the seeds of *O. violaceus* possess a high oil content, primarily consisting of beneficial unsaturated fatty acids. This quality makes them highly valuable as oil plants, offering potential health benefits and disease prevention [8]. Moreover, the seeds of *O. violaceus* are considered to have significant edible value.

It has been established in modern pharmacological studies that the natural active ingredients in *O. violaceus* have a wide range of pharmacological activities, including antibacterial, antitumor, antiradiation, hepatic protection, and other physiological activities [9,10,11]. Based on domestic and foreign research, flavonoids [10], alkaloids [12], phenylpropanoids [13], phenolic acids [14], and terpenoids [15] have been identified as the main components of *O. violaceus*. Despite being present in relatively small amounts, *O. violaceus’s* steroid, fatty acid, protein, and vitamin content are crucial to the plant’s overall value.

Recently, the scope of utilization of the plant has been expanding from an ornamental to an edible plant. *O. violaceus* seeds have high levels of unsaturated fatty acids and low levels of erucic acid, making them an outstanding quality oil resource in terms of economic value. The amino acid content of *O. violaceus* is well known and it can be used as feed in conjunction with maize and sorghum [16]. On the other hand, *O. violaceus,* as a wild vegetable in folk, also has a long-term edible history. People in Jiangsu and Anhui have been harvesting and eating the tender stems and leaves of *O. violaceus* for a long time. *O. violaceus* is rich in vitamin C, carotene, and so on, which has potential edible and medicinal value for the human body to prevent cold, cancer, and cardiovascular and cerebrovascular diseases.

Based on a review and summary of the literature, this paper comprehensively summarized and analyzed the research progress in recent years in terms of nutritional composition, chemical composition, and biological activity, intending to provide guidance and reference for the subsequent research and application of *O. violaceus*.

## 2. Research Methodology

We looked up information on the nutrients, chemicals, and health effects of *O. violaceus* by searching through databases like Elsevier, Journal of the American Chemical Society (ACS), PubMed, Web of Science, and China National Knowledge Infrastructure (CNKI). We searched for information in both Chinese and English. The keywords used included related words such as *O. violaceus*, Er-yue-lan, alkaloid, flavone, chemical composition, pharmacology, nutritional composition, application, and so on.

## 3. Nutritional Value

### 3.1. Vitamins

Vitamins are a type of small organic compounds that are essential for keeping our bodies and animals functioning properly. They are only needed in small amounts but play a crucial role in human growth, metabolism, and overall development. As the body cannot synthesize vitamins or does so in small quantities, people can only consume them from food.

Li Xinhua et al. [17] compared the vitamin content of wild *O. violaceus* with that of the cultivar Brassica chinensis at the seedling and moss stages. The levels of VC (vitamin C), VE (vitamin E), VB1 (vitamin B1), and VB2 (vitamin B2) were measured using the fluorometric method. On the other hand, the levels of VK (vitamin K), VB6 (vitamin B6), VB12 (vitamin B12), VB3 (vitamin B3), and β-carotene were determined using HPLC (high-performance liquid chromatography). The results are depicted in Table 1. Luo Peng et al. [18] also compared the major vitamins of *O. violaceus* with other plants of the genus Brassica, where β-carotene was determined by paper chromatography, which is shown in Table 2. Rich in vitamin C and carotenoids, the major vitamin content of *O. violaceus* is equal to that of common greens and even higher than greens in some cases. The homeostasis of vitamins in the human body is particularly important for health. Some studies have illustrated the importance of vitamin C in early infant development, especially in the brain and cognition [19]. In addition, vitamin C deficiency can have a negative impact on infectious diseases, cancer, diabetes, sepsis, and cardiovascular disease, among others [20]. Furthermore, β-carotene has been shown to significantly reduce the risk of oral, laryngeal, and breast cancers [21]. *O. violaceus* has a higher overall nutritional value compared to other vegetables in the Brassica genus, so it has great potential application prospects.

### 3.2. Amino Acid

Amino acids are the fundamental components of life. They serve as the building blocks for proteins and enzymes. They belong to a group of organic compounds that contain both amino and carboxyl groups within their molecules. The regulation of substance metabolism and information transmission in the body are two critical functions of amino acids, which are crucial for the nutrition, survival, and development of all life forms. As macromolecules, amino acids have a crucial role to play in the activity and physiological functions of biomacromolecules; as ligands, they coordinate with a variety of metal ions and offer information for the study of antitumor and anticancer drugs.

Amino acids can be divided into two categories: essential amino acids and nonessential amino acids. About 60% of amino acids can be synthesized in the liver, while the remaining 40% cannot be synthesized in the body and need to be obtained from food, which are called essential amino acids.

Weng Debao [22] determined the content of common amino acids in wild *O. violaceus*, as shown in Table 3. By using the method of fuzzy discernment and ratio coefficient of amino acids, using egg protein as standard protein, comparing with legume vegetables and wild vegetables, it was found that the protein content of fresh *O. violaceus* at the seedling stage was 3.01% and the amino acid content of the stem and leaf accounted for 89.98% of the crude protein. The results showed that the protein quality of *O. violaceus* was far superior to the contrast.

The variety of amino acids in *O. violaceus* is very rich. In addition to the eight essential amino acids, two uncommon amino acids, ornithine, and γ-aminobutyric acid, were also detected by Li Xinhua et al. [17].

Weng Debao et al. [23] used the Kjeldner method to determine the total protein content of *O. violaceus* and obtained the optimal extraction process conditions of the total protein of *O. violaceus* through single-factor experiments and response surface designs. Under the optimal extraction conditions, the protein extraction efficiency of *O. violaceus*’s leaf was 24.40 g/100 g.

### 3.3. Mineral Element

A variety of mineral elements are present in *O. violaceus*. Cao Weidong et al. [24] conducted measurements of macroelements, secondary elements, heavy metal elements, trace beneficial elements, and rare earth elements in different organs of *O. violaceus*, and the results are shown in Table 4.

### 3.4. Fatty Acids

As a new type of oil plant resource with outstanding quality, the seed of *O. violaceus* has a high oil content that can reach 46.91%, which is suitable for oil extraction. *O. violaceus* seeds have a fatty acid profile that is extremely low in erucic acid and high in nutrient-dense unsaturated fatty acids like linoleic and linolenic acids [25,26].

Zhou Min [25] compared three approaches to extracting *O. violaceus* seed oil in order of oil yield: ultrasonic-assisted method > hydroenzymatic method > physical pressing method.

Zhao Ru et al. [27] microwave-mediated immiscible binary solvent extraction (MIBSE) to extract *O. violaceus* seed oil and epigoitrin simultaneously and optimized by response surface method. Under the optimal extraction conditions, the yield of *O. violaceus* seed oil by MIBSE was 34.08% ± 1.38% and that by conventional Soxhlet extraction (SE) method was 25.50% ± 2.63%. The oil yield of MIBSE was 1.34 times higher than that of the SE method. There were 11 kinds of fatty acids in *O. violaceus* seed oil, including 5 saturated fatty acids (SAFAs, C_n:0_, 17.73%), 3 kinds of monounsaturated fatty acids (MUFAs, C_n:1_, 22.12%), and 3 polyunsaturated fatty acids (PUFAs, C_n:2,3_, 60.15%). The content of unsaturated fatty acids was much greater than that of saturated fatty acids and the fatty acid composition is as follows: Table 5.

Arachidic acid, arachidonic acid, and a small amount of erucic acid were found in *O. violaceus* by Sun Xiaoqin et al. [28]. Xue Yinghong et al. [29] also found that the total fatty acid content of *O. violaceus* seed increased gradually with increasing seed development, with the highest total fatty acid content in its late stage. Lv Zhongjin [30] compared *O. violaceus* with two types of oilseed rape and found that the linoleic acid content of *O. violaceus* seed oil was higher than that of both kale-type oilseed rape (Ning-You-7-Hao) and Canadian oilseed rape (Westar) and the erucic acid content was lower than either of these.

Liu Sifan et al. [31] isolated β-D-glucopyranosyl-12-hydroxyjasmonate, a derivative of fatty acid, from *O. violaceus* seed with a structure as shown in Figure 1.

Due to its high oil content, low acid value, and favorable fatty acid ratio, the seed oil of *O. violaceus* holds great potential as an edible oil. It shows promising prospects for development in the food industry. High-quality oil plants with low erucic acid have been cultivated by cross-breeding of *O. violaceus* and kale-type oilseed rape [32].

In addition, *O. violaceus* is also rich in 24-carbon dihydroxy fatty acids (diOH-FAs) composed of triacylglycerol (tAG) estolides, such as Nebraska acids and Wuhan acids, which have better high-temperature resistance and lubrication properties than castor oil and are considered as a potential industrial oil crop [33,34,35].

### 3.5. Others

The utilization value of the seed cake of *Brassicaceae* depends mainly on its contents of crude proteins and toxic glucosinolates. In a study conducted by Lv Zhong jin [30], *O. violaceus* seed cake was compared to two types of rape (rapeseed). According to the data in Table 6, it was observed that the protein content in *O. violaceus* seed cake was remarkably high, reaching 52.32%. This surpassed the protein content found in typical rapeseed cake. Its glucosinolate content was similar to that of normal rapeseed cake.

## 4. Chemical Compositions

### 4.1. Flavonoids

Flavonoids are a huge class of naturally occurring substances that are found throughout nature. They are mostly found in plants’ stems, leaves, flowers, and fruit parts and are coupled with sugars to make glycosides [36]. In the available reports, flavonoids account for the majority of the compounds in *O. violaceus*. Based on a consolidation of the literature, it is clear that flavonoids are also the main active site in *O. violaceus*. Flavonoids have a variety of biological activities such as antibacterial, antiviral, antioxidant, anti-inflammatory, and antitumor.

The ethanol extract of the stem and leaves of *O. violaceus* in a study by Weng Debao et al. [37] reacted positively with various color development reagents, indicating that the stem and leaves of *O. violaceus* contained a variety of flavonoids. The results showed that the ethanol extract contained quercetin, kaempferol, and isorhamnetin, and the total flavonoid glycoside was 0.568%. Wang Xin et al. [38,39] investigated the flavonoid content of different extraction methods of *O. violaceus* as follows: microwave-assisted 70% ethanol extraction > 70% cold ethanol extraction > water extraction; the flavonoids of *O. violaceus* were specifically tested for using the ADS-17 resin as an adsorbent. After purification, the purity of flavonoids reached 58%, which was increased by about 6.5 times. Sai Chunmei et al. [40] found that the total flavonoid content of different parts of *O. violaceus*: leaf > flower > stem > root > fruit.

In the existing research on *O. violaceus* chemical composition, it is found that there are many kinds of flavonoids. Based on reading a large number of pieces of the literature, after sorting and classifying a variety of flavonoids, all the reported flavonoids are now divided into flavanols (1~12), flavanones (13~16), isoflavones (17), flavones (18~22), and flavan-3-ols (23). Flavonoids have been reported in *O. violaceus* as shown in Figure 2 and Table 7.

### 4.2. Alkaloids

*O. violaceus* also has a high concentration of alkaloids, which are alkaline chemicals with nitrogen heterocyclic rings and optical activity. Most alkaloids have biological activities such as antibacterial, antiviral, and anti-inflammatory, and are often the effective ingredients in many Chinese herbs and medicinal plants, which have great potential in healthcare [44,45].

Zhu Nai Liang et al. [12] obtained eight new alkaloids: Orychophragmuspine A–H, from the aqueous extract of the seeds of *O. violaceus* through the 10% ethanol part of D101 resin; Liu Chenqi et al. [46] also obtained a new alkaloid compound Orychophragmuspine I from the 10% ethanol part of the seed D101 resin; Zhang Guangjie et al. [47,48] isolated four new alkaloids: Orchophragine A–D from the n-butanol extract fraction of the ethanolic extract of *O. violaceus* seeds via the 50% ethanolic aqueous part of AB-8 resin. Xia Xinyi et al. [49] derived epigoitrin from aqueous decoction of *O. violaceus* seeds in the range of 0.64~0.92%.

The alkaloid compounds reported in *O. violaceus* are shown in Figure 3 and Table 8.

### 4.3. Phenylpropanoids

Phenylpropanoids are typically classified into three constituent groups: phenyl propionic acids, coumarin, and lignans.

Coumarin is a general term for a class of compounds with a benzo-α-pyranone parent core, which is widely found in various plants in nature and also in a few fungi and bacteria. Studies have shown that coumarins have a wide range of pharmacological activities, such as anti-inflammatory, antioxidation, antivirus, antitumor, anticoagulation, and antibacterial effects [54,55]. The oxidative polymerization of phenylpropanoids, which are frequently found in plant resins or xylem, results in the formation of lignans, a class of natural products.

In published studies [13], three new iso-coumarins and one new dihydroisocoumarin were isolated from the seeds of *O. violaceus*: orychophramarin A–D.

The reported phenylpropanoids from *O. violaceus* are shown in Figure 4 and Table 9.

### 4.4. Phenolic Acid

Phenolic acids are natural compounds with a polyphenolic structure and are also important active ingredients for medicinal plants to function, with pharmacological activities such as antitumor, antioxidant, anti-inflammatory, anticoagulant, antiatherosclerosis, and lowering blood lipids, etc. Chinese herbs such as *Angelica Sinensis* and *Salvia miltiorrhiza* are rich in phenolic acids [56,57,58].

The *O. violaceus* water extract contains a significant number of phenolic acids. Xia Xinyi [14] isolated 15 compounds, including salicylic acid and p-hydroxycinnamic acid, from the aqueous extracts of *O. violaceus* using a macroporous resin column.

The phenolic acids reported in *O. violaceus* are shown in Figure 5 and Table 10.

### 4.5. Terpenoids

Terpenoids are the most abundant group of natural products with complex skeletons and diverse biological activities, including antibacterial, antiallergic, antitumor, anti-inflammatory, vasodilating, and blood-glucose- and lipid-regulating properties [59,60,61].

In addition to its content of triterpenes and their saponins, *O. violaceus* is known to possess various terpenoids. Studies have revealed the presence of a novel triterpene saponin, along with ten previously identified triterpene saponins, isolated from the seeds of *O. violaceus* [15,41]. The reported terpenoids of *O. violaceus* are shown in Figure 6 and Table 11.

### 4.6. Steroids

Steroid compounds are widely available as a natural chemical constituent and all contain a cyclopentane and polyhydrophenanthrene steroid nucleus in their structure. Steroids are capable of performing important physiological functions with antitumor, anti-inflammatory, cerebral-blood-flow-improving, and cerebrovascular-resistance-reducing effects [62,63,64]. Common steroidal drugs, such as steroid hormones and cardenolide, play an important role in medical applications.

Steroids have been reported for *O. violaceus*, see Figure 7 and Table 12.

### 4.7. Anthocyanidin

Anthocyanidins are a kind of compound based on the flavonoid nucleus, which is widely found in the cytosol of flowers, fruits, stems, leaves, and other organs of plants and has activities such as eliminating free radicals in the body, anti-inflammatory, and antitumor. Anthocyanins are a class of compounds that combine anthocyanidins with sugars.

Toshio Honda et al. [65] extracted three acylated cyanidin 3-sambubioside-5-glucosides (Orychophragonus violet-blue anthocyanins, OVAs) from the petals of *O. violaceus*, cyanidin3-[2-(2-(4-(6-(4-glucosylcaffeoyl)-glucosyl)-caffeoyl)-xylosyl)-6-(4-glucosyl-p-coumaroyl)gIucoside]-5-gjucoside,cyanidin3-[2-(2-(4-(6-(4-glucosylcaffeoyl)-glucosyl)-caffeoyl)-xylosyl)-6-(4-glucosyl-feruloyl)glucoside]-5-glucoside and cyanidin3-[2-(2-(4-(6-(4-glucosylcaffeoyl)-glucosyI)-caffeoyl)-xylosyl)-6-(4-glucosyl-sinapoyl)glucoside]-5-(6-malonylglucoside). A study [66] demonstrated that anthocyanins derived from the petals of *O. violaceus* have the potential to serve as an alternative to phenolphthalein in monitoring the carbonation process in cement paste and evaluating the long-term strength and resilience of concrete.

Based on a study [67], it was found that both pink and blue-purple petals of *C. chinensis* contained a total of seven anthocyanin components. Among these, six were identified with chemical structures presented in Figure 8 and Table 13. Interestingly, white petals of the same species did not have any anthocyanins and none of the colors (pink, blue purple, and white) contained carotenoids.

## 5. Pharmacological Activity

### 5.1. Antioxidant Properties

Many human diseases, such as tumors, inflammation, cancer, coronary heart disease, aging, and atherosclerosis, are caused by free radicals that are not removed in time and continue to react to generate reactive oxygen species (ROS). ROS can react with almost all biological macromolecules in various types, causing a series of harmful effects [68].

Flavonoids exhibit excellent free radical scavenging and antioxidant capacity, blocking the production of free radicals in the body. The mechanism of action includes a reaction with O^2^-to block the free radical chain reaction, chelation with metal ions to block free radical production, and reaction with lipid peroxides (ROO) to block the lipid peroxidation process [69,70].

Quercetin and catechins, two well-known flavonoids, are extensively found in various plant species. These compounds exhibit a synergistic effect in protecting HepG2 cells from oxidative stress damage induced by H_2_O_2_. Quercetin and catechins concertedly enhanced HepG2 cell survival and catalase (CAT), superoxide dismutase (SOD), and glutathione peroxidase (GSH-Px) activities and cooperatively inhibited cellular ROS production, MDA accumulation, and apoptosis [71].

It has been reported [42] that 3,4′,6,8-tetyahydroxy-flavone-7-C-glucoside, 2,3-dihydroxybenzoic acid, quercetin 3-*O*-β-d-glucopyranosyl-7-*O*-α-L-rhamnoside, Kaempferol-7-*O*-α-L-rhamnoside, and quercetin isolated from the whole herb of *O. violaceus* have antioxidant capacity. Among them, 3,4′,6,8-tetyahydroxy-flavone-7-C-glucoside, 2,3-dihydroxybenzoic acid, and quercetin had stronger antioxidant capacity than the positive control drug Trolox.

The autooxidation of lard can be significantly delayed by using the ethanol extract of *O. violaceus* and its antioxidant activity increases with concentration. Flavonoids can provide active hydrogen protons, thus blocking lipid peroxidation [72]. The total flavonoid extract of *O. violaceus* has a scavenging effect on OH produced by the Fenton system. It is known that flavonoids in *O. violaceus* have antioxidant activity and the higher the flavonoid content, the stronger the antioxidant capacity [38,39].

Using the oxidative stress injury model of HepG2 cells induced by H_2_O_2_, it was found that orychophragmuspine I pretreatment increased the cell survival rate, suggesting that orychophragmuspine I has a significant protective effect on cell damage caused by H_2_O_2_. In this study, antioxidant biochemical parameters were also measured to verify that orychophragmuspine I could inhibit the leakage of LDH, inhibit the increase in MDA level, and enhance the activity of SOD and GSH-Px. LDH is leaked from cells during cell membrane damage and can be used as a marker of cell damage; MDA, as a degradation product of cell membrane lipid peroxidation, is an essential marker of oxidative damage; SOD and GSH-Px both have functions of resisting oxidative damage. Western blot analysis showed that orychophragmuspine I enhanced Nrf2, HO-1, and GCL expression, revealing that orychophragmuspine I may mediate antioxidant function by activating the Nrf2-ARE pathway to reduce cell damage caused by H_2_O_2_ [46].

Zhu Nai-Liang et al. [12] demonstrated that orychophragmuspine A could inhibit H_2_O_2_-induced oxidative stress and LDH leakage indirectly or directly by reducing ROS levels and enhancing SOD activity, validating the antioxidant effect of orychophragmuspine A.

### 5.2. Antiradiation Properties

Ionizing radiation causes cell damage and apoptosis, leading to various related diseases. It acts directly on biological macromolecules and indirectly affects water molecules in living organisms [73]. Natural active ingredients in plants and animals can provide protection or repair against radiation damage through various effects such as antiemetic, anti-inflammatory, antioxidant, cell proliferation, and wound healing [74]. At present, it has become a research hotspot to explore the active ingredients of traditional Chinese medicine. Among them, the common alkaloids with antiradiation damage effects mainly include tetramethylpyrazine and matrine [75].

Zhang Guangjie et al. [41,48] used an 8 Gy dose of ^60^Coγ radiation to induce a radiation model of HUVEC cells and found that orychophragine C and orychophragine D significantly increased the survival rate of HUVECs after irradiation. Treatment with orychophragine D markedly enhanced the survival rate of male C57/BL mice exposed to 8Gy(lethal dose) ^60^Coγ radiation in a dose-dependent manner; from 10 to 14 days after irradiation, the leukocyte, erythrocyte, hemoglobin, and platelet counts of mice in the orychophrine-D-treated group were significantly higher than those of the control group, indicating that orychophrine D can reduce radiation-induced cell damage, accelerate the progress of cell repair, and alleviate radiation-induced hematopoietic toxicity. It has a good radiation protective effect.

Xu Zanxin et al. [51] induced HUVEC by irradiation with ^60^Coγ ray at a dose of 8Gy and determined by the CCK-8 method, showing that (−)-orychoviolines A and (±)-orychoviolines A at a concentration of 25 μM respectively increased the viability of irradiated HUVEC cells from (52.4% ± 2.3%) to (66.4% ± 2.8%) and (62.2% ± 2.6%). Based on the findings from the comet assay, it was proposed that (−)-orychoviolines A may exert a radioprotective effect by safeguarding the DNA of HUVEC cells.

Phenylacetamide and uridine, both isolated from the whole herb of *O. Violaceus*, exerted antiradiation effects, which raised the survival of irradiated HUVEC cells from (38.7% ± 1.44%) to (56.2% ± 2.66%) and (54.4% ± 2.89%) at a concentration of 25 μM [42].

### 5.3. Antineoplastic Properties

Malignant tumors have gradually become one of the major diseases threatening human health due to their biological characteristics, such as cell differentiation, abnormal proliferation, and metastasis, with increasing morbidity and mortality. Currently, surgery, radiotherapy, and chemotherapy are the standard methods for treating cancer but these treatments frequently have serious side effects. Contemporary research has found that the active ingredients of traditional Chinese medicine can effectively inhibit tumor growth, reduce tumor recurrence rates, and improve patients’ quality of life and survival time, and have few toxic side effects [76]. Chinese medicine has two main types of antitumor effects: one is to exert direct inhibitory effects on the growth of tumor cells, such as cytotoxic effects and induction of apoptosis; the other is to exert indirect antitumor effects by improving the pathological state of the body and thus activating the immune response of tumor cells, alleviating complications and inhibiting the migration of tumor cells [77,78]

Zhang Guangjie et al. [13] used Hela and HCT-116 cells as in vitro tumor models and found that orychophramarin A, orychophramarin B, and orychophramarin C all had the activity of inhibiting the growth and proliferation of tumor cells in vitro. In particular, orychophramarin A showed a significant inhibitory effect on the above two cell lines, whose IC50 values, respectively, were 8.91 and 5.10 μM, which surpassed the efficacy of the positive drug cisplatin. The coumarins orychophramarin A–C showed better activity than the dihydroisocoumarins orychophramarin D in inhibiting the growth of tumor cells. The activities of orychophramarin A and C were higher than those of orychophramarin B, presuming that the 8-OH in the structure was the synergist structure by the structural-activity relationship analysis while orychophramarin A was more active because of the 7,8-hydroxyl substitution in the ring.

It was found that orychophramarin A blocked the HCT-116 cell cycle at the G2 phase in a dose-dependent manner. The DNA content of the G2 phase was 14.51% in the blank control group and was 20.77%, 27.86%, and 36.36% when the drug concentrations were 3.13, 6.25, and 12.5 μM, respectively (Table 14).

In early apoptosis, phosphotidylserine can flip from being primarily distributed on the inside of the cell membrane to the outside. Flow cytometry analysis showed that orychophramarin A could significantly promote the apoptosis of HCT-116 cells in a dose-dependent manner. The apoptosis rate was (8.22% ± 0.53%) in the blank control and after treatment with 3.13, 6.25, and 12.5 μM orychophramarin A, the apoptosis rates, respectively, were (13.47% ± 2.42%), (25.03% ± 3.86%), and (35.43% ± 3.39%). This finding suggests that orychophramarin A exerts its antitumor activity by preventing tumor cells from growing and multiplying by directly inducing apoptosis.

Orychoside A, mimengoside C, mimengoside D, buddlejasaponin I, and buddl-ejasaponin III demonstrated potent inhibition of tumor cell growth in vitro. These compounds exhibited significant inhibitory effects on HepG2 and A549 cells, with IC50 values ranging from 0.48 to 11.63 μM. Among them, buddlejasaponin I and III have the most significant activity. A conformational study of these compounds showed that HepG2 and A549 cells are sensitive to triterpenoid saponins whose glycosides are saikogenin. When 28-OH forms an intramolecular ether bond with C-13, its tumor cell suppressive activity is remarkably enhanced [41].

### 5.4. Hepaticprotective Properties

Hepatic damage can be caused by viruses, drugs, alcohol, and trauma, and liver disease has become one of the most common diseases worldwide. Alanine aminotransferase (ALT), glutamic oxalacetic transaminase (AST), and total bilirubin (TBiL) in serum can reflect the damage to hepatocytes. Elevated levels of ALT, AST, and TBiL indicate severe damage to liver cells. In addition, MDA (malondialdehyde) often plays a role in apoptotic necrosis, often in conjunction with cytokines released from hepatocytes [79,80].

Thioacetamide (TAA), a common industrial bactericide, is hepatotoxic to the liver and can be metabolized into sulphone and sulfoxide in the body, producing large amounts of free radicals and ROS, causing hepatocyte necrosis, which leads to the entry of ALT and AST from the cytoplasm into the blood, ultimately increasing the levels of ALT and AST in the serum [81]. Peng Min et al. [82] used male ICR mice to establish a TAA-induced acute liver injury model. The results showed that compared to the model group, MDA content in the liver tissue in the high-dose group of aqueous extract of *O. violaceus* (AEOV) decreased by 52.4%, and SOD, GSH-Px, and GSH increased by 42.01%, 38.8%, and 136.41%, respectively. In addition, compared to the model group, the activities of ALT and AST and the content of TBiL in the serum of the high-dose AEOV group, respectively, decreased by 84.57%, 87.62%, and 10.26%, confirming the protective effect of AEOV on the liver. In terms of the protective effect on TAA-induced liver injury, it has been observed that AEOV can decrease the levels of NF-κB p65 and keap-1 proteins in liver tissue while increasing the content of Nrf2 protein. This indicates that AEOV promotes the activation of the Nrf2 pathway. Additionally, AEOV inhibits the phosphorylation of p38 and ERK, effectively suppressing the activation of the NF-κB-signaling pathway and reducing the occurrence of inflammation. By inhibiting the phosphorylation of p38 and ERK, the activation of the NF-κB-signaling pathway and the occurrence of inflammation are suppressed. By controlling the phosphorylation of JNK, it can increase the ratio of Bcl-2/Bax, restrain the Caspase-3 conversion into Cleaved Caspase-3, and curb the apoptosis of hepatocytes, thus playing a role in liver protection.

Acetaminophen (APAP) is a widely used analgesic-antipyretic. Prolonged or large doses of APAP produce a large accumulation of the toxic metabolite NAPQI, which causes oxidative stress and mitochondrial damage by depleting glutathione and producing ROS, eventually leading to hepatocyte apoptosis. Mitochondrial oxidative stress has been implicated as the key to APAP-induced liver injury and the caspase pathway by which APAP mediates apoptosis is essential for its contribution to acute liver injury [83]. In a study conducted by Pang Min et al. [84], a model of APAP-induced acute liver injury was established using male C57BL/6J mice. The researchers discovered that the high-dose group of AEOV exhibited a significant decrease of 29.69% in MDA content in the liver homogenate compared to the model group. Additionally, they observed a substantial increase of 48.68% in SOD levels and a remarkable increase of 232.80% in GSH levels. These findings indicate that the aqueous extract of *O. violaceus* effectively enhances the antioxidant capacity in the liver. Compared to the model group, the ALT and AST activities of the AEOV high-dose group, respectively, decreased by 71.38% and 59.33%, indicating that the aqueous extract of *O. violaceus* had a significant effect on hepatic protection; the AEOV high-dose group could significantly improve the liver swelling caused by APAP in mice and the morphology of mouse liver cells improved obviously. Based on the mechanism of its protective effect on APAP-induced liver injury, it was shown that AEOV could promote the phosphorylation of AMPK, AKT, and GSK-3β protein expression, activate the AMPK/AKT/GSK-3β signaling pathway and inhibit Caspase-3 expression; it could also inhibit JNK phosphorylation, up-regulate Bcl-2 protein expression and down-regulate Bax and Caspase-9 protein expression, thus inhibiting apoptosis and exerting a protective effect on APAP-induced liver injury in mice.

ConA, which is a T cell mitogen, can activate T lymphocytes and induce excessive immune responses. Simultaneously, it stimulates the body to secrete significant amounts of inflammatory factors such as INF-γ, TNF-α, IL-1, IL-2, and IL-6. Under the mediation of these inflammatory factors, it causes an autoimmune inflammatory response, which leads to liver damage. ALT and AST will enter the blood in large amounts from hepatocytes and their concentrations increase with increasing doses of ConA [85,86]. Using male BALB/C mice to compose a model of acute liver injury induced by ConA, Pang Min [87] showed that the high-dose group of AEOV could significantly reduce ALT and AST activity levels, enhance SOD and CAT activities and increase GSH content, and reduce MDA content by 36.70% compared to the model group, indicating that AEOV could increase the antioxidant capacity and improve the damage of hepatocytes in mice caused by ConA.

The liver serves as the primary metabolic organ for alcohol and oxidative stress triggered by alcoholism in the liver is considered a crucial mechanism behind alcoholic liver injury. Ethanol is metabolized into acetaldehyde in liver cells, during which a large number of ROS will be produced, thus promoting oxidative stress of liver cells, leading to liver damage and inflammatory response, resulting in a series of changes such as liver degeneration [88,89]. Liu Chenqi et al. [90] established a model of acute alcoholic liver injury in ICR mice by giving 56° liquor. The results of the experiment showed that AEOV was able to antagonize the increased hepatic metabolic enzymes ALT, AST, and triglycerides (TG) caused by excessive alcohol and had a protective effect on the liver, whereas the positive drug bifendate did not have a significant effect on TG. AEOV significantly boosted the activity of SOD, ADH, and the levels of GSH-Px and GSH in the liver of mice with alcoholic liver injury and its intensity was comparable to that of the positive drug bifendate. The conclusion drawn suggests that AEOV may exhibit hepatoprotective effects by acting through the antioxidant damage pathway. Additionally, it is believed to inhibit lymphocyte infiltration as well as prevent hepatocyte apoptosis or necrosis.

Studies have shown that cortex dictamni can induce liver damage in animals. Furthermore, its main component, dictamin, has been found to have a toxic effect on the HepG2 cell line. In the case of prolonged consumption of aqueous decoction of the cortex, dictamni alone can cause significant liver damage [91]. By using cortex dictamni aqueous decoction and ICR mice to establish a mouse model of acute liver injury, Yewei Zhan et al. [92] measured the biochemical indexes of each group of mice and found that the levels of GPT, GOT, and LDH decreased by 99%, 98%, and 96%, respectively, in the AEOV high-dose group compared to the model group, indicating that AEOV could significantly alleviate the acute liver injury caused by cortex dictamni. The elevated MDA content in liver tissue and the decreased reduced GSH/GSSG ratio observed in the model group, when compared to the blank control group, indicate that the liver injury caused by cortex dictamni may be attributed to oxidative stress damage. The high-dose group of AEOV showed a 22% decrease in MDA and a 54% increase in GSH/GSSG ratio compared to the cortex dictamni model group, confirming the protective effect of AEOV against acute hepatic injury induced by cortex dictamni in mice through antioxidative stress effect.

Cocaine, which is highly addictive, can cause psychological dependence within a very short period after use, and long-term use can lead to languor. Other than the psychoactive effects mentioned above, cocaine causes other damage to the body, such as cardiovascular, gastrointestinal, muscular, and skeletal [93]. Cocaine is hepatotoxic to mice, rats, and humans. The oxidative metabolism of cocaine in the liver, facilitated by P-450 enzymes, plays a significant role in its hepatotoxicity. This metabolic process converts cocaine into highly toxic intermediates and generates a substantial amount of reactive oxygen species (ROS). As a result, lipid peroxidation occurs, particularly in cell membranes, leading to hepatocyte necrosis [94]. Xu Ziqian et al. [95] used cocaine hydrochloride to establish a model of acute liver injury in ICR mice. Compared to the model group, the GPT, GOT, and LDH activities of mice in the AEOV group decreased in a dose-dependent manner, indicating a reduction in the degree of liver injury after prophylactic administration of AEOV. This suggests that AEOV exhibits hepatoprotective effects against toxic chemicals, safeguarding the liver from their harmful impacts. The content of MDA in the liver tissue of the AEOV group was noticeably decreased compared to that of the model group, while the activity of CAT was substantially increased, which indicated that AEOV protected the liver cell membrane by antilipid peroxidation to achieve a protective effect against liver injury.

CCl_4_ activates hepatic Nrf2 protein and downstream gene expression, causing hepatocyte necrosis and fatty degeneration of the liver and long-term exposure can also induce liver fibrosis, resulting in hepatorenal toxicity [96,97]. Huo Xiaowei et al. [43] established a CCl4-induced liver injury model of BALB/C mice. When compared to the model group, mice in the AEOV group had lower serum levels of ALT and AST, higher levels of T-SOD, CAT, Gsh-PX, and GSH, and improved histopathological changes to the liver. AEOV ameliorates CCl4-induced liver injury by mediating Nrf2 to attenuate oxidative stress in the liver, reducing IKKα expression, and inhibiting p-IκBα protein phosphorylation to mediate the NF-κB pathway to inhibit inflammatory responses and TNF-α production.

Against H_2_O_2_-induced HepG2 cells, the epigoitrin isolated from the AEOV is protective [43]. Pretreatment of HepG2 cells with epigoitrin improved HepG2 cell viability, reduced LDH activity and MDA content, and elevated SOD and GSH-Px activity, while Nrf2, GCL, p-IκBα, and NF-κB expression were down-regulated, revealing that epigoitrin may have anti-inflammatory and antioxidant effects through regulating Nrf2 and NF-κB pathways, thereby reducing liver injury. Figure 9 illustrates the primary mechanism underlying the hepatoprotective effect of *O. violaceus* extracts.

### 5.5. Antiferroptosis Properties

Ferroptosis is a pattern of iron-dependent regulated cell death, closely related to dysregulation of iron metabolism, accumulation of ROS, lipid peroxidation, and inactivation of GPX4. Radiation produces large amounts of ROS, indirectly causing cell damage and inducing ferroptosis in cells [98].

Xiao Fengjun et al. [99] used Erastin to induce ferroptosis in IEC6 cells and found that orychophragine D had a ferroptosis inhibitory effect and was more potent than the positive drug Fer-1 at a concentration of 1 μM. In a radiation-induced ferroptosis cell model, the ferroptosis inhibitory activity of orychophragine D was also better than that of the positive drug Fer-1. Orychophragine D ameliorates pathological structural changes in the small intestinal crypts of mice after radiation exposure.

### 5.6. Anti-Inflammatory Properties

Inflammation is a defensive response of the body to stimulation by inflammatory mediators, usually manifested as redness, heat, swelling, and pain. Anti-inflammatory alkaloids generally have a ring structure with the nitrogen atom located within the ring, which can form salts with acids [100].

(−)-Orychoviolines A exhibits anti-inflammatory activity and displays a notable inhibitory effect on the release of NO, with an IC50 value of 20.3 ± 1.58 μM, which is comparable to that of the positive drug dexamethasone. Perlolyrine also demonstrates anti-inflammatory activity by inhibiting the release of NO, with an IC50 value of 27.3 ± 1.29 μM [50]

### 5.7. Antibacterial Properties

A study [101] isolated a protein, Zp, from the seeds of *O. violaceus*. Zp protein has an inhibitory effect on a variety of plant pathogenic fungi and maybe a kind of plant defensin protein. The semi-inhibitory concentration (IC50) for *Fusarium oxysporum* growth and spore germination was 54.36 μg/mL and 15.97 μg/mL, respectively.

## 6. Toxicity

ICR mice were subjected to acute oral toxicity and subacute toxicity tests using the limited dose method. The mice in each group grew and developed well, without poisoning or death, and their behavior and urine and feces were normal. The pathological histological examination of the organs of the mice in all groups had no obvious pathological changes, which proved that *O. violaceus* did not have any obvious acute toxicity and short-term toxic side effects on the test animals and was a practically nontoxic substance [102].

## 7. Application Value

*O. violaceus* is abundantly distributed in our country, with ample resources. It is known for its resilience to adverse conditions, including cold temperatures, and it is rich in diverse nutrients. It is an excellent ground cover plant that integrates ornamental and ecological benefits, feeding, edible, and medicinal plants, whose seeds can be used as excellent oil materials.

### 7.1. Ornamental Value

It blooms in early spring and has a long flowering period, generally lasting from March to June. The inflorescences are from bottom to top and the petals change gradually from white to purple in a transitional gradient. A few of them are pure white or pure purple. When cultivated on a large scale, *O. violaceus* exhibits a captivating sight during the full flowering stage, with a vast expanse of pale-blue-purple flowers creating a stunning and picturesque scenery [103].

*O. violaceus* is a rare ground cover plant that flowers in early spring and is green in winter in northern China, which has strong environmental adaptability and tolerance to cold, shade, and drought and can grow in grasslands, plains, hillsides, roadsides, and shady mountains [104].

### 7.2. Ecological Value

*O. violaceus* benefits from early greening, a large number of branches, a large area of single leaves, and a developed root system that can quickly cover the ground. When planted at a large scale, *O. violaceus* can fix ground dust and effectively hinder surface dust; at the same time, its roots promote the formation of soil agglomerate structure, reducing the surface runoff to the storage water source and effectively preventing soil erosion. Using the photosynthesis of plants to absorb carbon dioxide and release oxygen, they also absorb and decompose harmful substances in the air and purify it [105].

Studies [106,107] have shown that, as a clean and sustainable maize cultivation mode, the *O. violaceus*-maize rotation can adjust the pH of calcareous soil, reduce the soil C/N ratio, improve maize yield, reduce the amount of fertilizer, and improve the utilization rate of fertilizer. At the same time, it can reduce greenhouse gas emissions and reactive nitrogen loss, ameliorate soil composition, lessen environmental damage, and achieve environmental and economic benefits.

### 7.3. Feeding Value

*O. violaceus* is rich in vitamins and proteins and has a high yield, with a higher variety and content of amino acids than ordinary fodder. It can be used as green fodder and can also be dried and made into compound fodder with corn and wheat bran as overwintering fodder grazing areas. Furthermore, the seed cake remaining after oil extraction from *O. violaceus* is rich in protein content. This seed cake can be processed into a high-protein concentrate feed, thus offering a novel and valuable source of high-quality vegetable protein [108].

### 7.4. Edible Value

*O. violaceus* can be eaten as a vegetable; its stems and leaves contain high levels of protein, calcium, iron, carotene, vitamin C, and other nutrients. Each 100 g of *O. violaceus* stems and leaves contains 3.32 mg of carotene, VB_2_0.16 mg, and 59 mg of VC, which is beneficial for both the treatment and prevention of illnesses like scurvy and night blindness. To get rid of the bitterness, the stems and leaves can be cooked in a stir-fry or added to soups [109].

Early spring sees consumption of the young leaves, late spring consumption of the moss, and autumn harvesting of the seeds. The seed of *O. violaceus* is a high-quality oil resource with an oil content of over 50%, which with high linoleic acid and low erucic acid and can be extracted into premium edible oil, and an important natural genetic resource for improving the quality of rape. Linoleic acid is not only easily digested and absorbed by the body but also has the function of lowering cholesterol and triglyceride and will soften blood vessels and prevent the formation of blood clots, which has therapeutic effects. *O. violaceus* can also be developed to make food products such as vegetable biscuits and drinks [108].

### 7.5. Medicinal Value

According to a description, “*O. violaceus* is warm-natured, pungent, bitter, and sweet in taste” and it is effective in treating jaundice in children and deep-rooted breast carbuncles. The whole herb is used as a medicine for regulating pneuma, harmonizing the stomach, and promoting diuresis and detoxification.

Modern research has confirmed the pharmacological effects of *O. violaceus*, including antioxidant, antiradiation, antitumor, hepatoprotective, antiferroptosis, anti-inflammatory, and antibacterial effects, and its health and medicinal values are increasingly being studied by scientists.

## 8. Conclusions and Future Perspectives

*O. violaceus*, which is abundant in resources and widely distributed, exhibits a range of pharmacological activities and has numerous potential uses in agriculture, the production of oil, food, and medicine. The seeds of *O. violaceus* have a high oil content with high linoleic acid and low erucic acid. In addition, as an edible vegetable, it is rich in many essential nutrients such as vitamins, proteins, and minerals. Among other things, it contains a variety of bioactive compounds such as flavonoids, alkaloids, phenylpropanoids, phenolic acids, terpenoids, and steroids that have been isolated and identified from *O. violaceus*. The extracts and the isolated compounds have been reported to have antioxidant, antiradiation, antitumor, hepatoprotective, antiferroptosis, anti-inflammatory, and antibacterial activities. Hence, it can be concluded that the development and utilization of *O. violaceus* in the fields of medicine and food are still considerably lacking and have yet to reach their full potential.

This paper provides an overview of the nutritional composition, chemical composition, bioactivity, and application value of *O. violaceus* and analyzes the correlation between its chemical composition and bioactivity (see Figure 10), providing a theoretical basis for in-depth research and product development of *O. violaceus*. To date, over 110 compounds have been identified in *O. violaceus*, most of which are flavonoids and alkaloids, as well as being the main bioactive compounds responsible for most of its pharmacological activity in several of the identified chemical constituents. The extensive literature has supported these ideas but more in-depth research is still required. Based on reports in the literature, a summary of their biological activities is provided in this paper (see Figure 11).

Most of the research on the pharmacological action of *O. violaceus* focuses on its aqueous and alcoholic extracts but there is a lack of studies on the biological activity of monomer compounds, leading to the uncertainty of the material basis of the pharmacological action of *O. violaceus* and the mechanism of pharmacological activity is even less. Currently, the *O. violaceus* liver protection effect serves as the main focus of its pharmacological mechanism. As well as reducing inflammation and apoptosis, it safeguards the liver from damage caused by antioxidative stress.

One of the most important factors in controlling the body’s antioxidative stress mechanism is the nuclear factor erythroid-2 related factor (Nrf2). Under normal conditions, Nrf2 exists in cells as a complex with the epoxy chloropropane Kelch sample-related protein-1 (Keap1), so Keap1 exerts a negative regulatory effect on regulating Nrf2 function [110,111]. When stimulated by external ROS or when Nrf2 is phosphorylated, Nrf2 is uncoupled from Keap1, and the activated Nrf2 translocates into the nucleus and binds to the ARE promoter, regulating downstream proteins such as phase II metabolic enzymes, antioxidant enzymes, and anti-inflammatory factors, thereby exerting an antioxidant effect [112] (see Figure 12). The antioxidant effect of *O. violaceus* is mainly mediated by the Keap1-Nrf2/ARE signaling pathway that regulates antioxidant proteins/enzymes including HO-1, GCL, SOD, GSH, GSH-Px, etc.

The NF-κB signaling pathway, an important regulator in the immune system, is a typical downstream inflammatory pathway, of which IKK/IκBα/NF-κB is one of the most classical pathways and is a common pathway for the production of various inflammatory factors [113]. In the resting state, NF-κB p65 is bound to IκBα and located in the cytoplasm. Upon stimulation by exogenous agents like LPS, the IκB kinase (IKK) complex phosphorylates IκBα. NF-κB p65 dissociates from phosphorylated IκBα and rapidly enters the nucleus and then binds to the κB site to promote the release of relevant inflammatory factors (IL-6, TNF-α, etc.), while cytokines further activate NF-κB, creating positive feedback regulation [114,115]. In addition, MDA can combine with proteins to form malondialdehyde-acetaldehyde adducts (MAA) that upregulate inflammatory factors and induce inflammation [116]. *O. violaceus* can decrease MDA content, reduce the phosphorylation of p38 and ERK protein, significantly down-regulate the expression of phosphorylated protein of IκBα and NF-κB p65, inhibit the activation of NF-κB signaling pathway, and thus play a role in inhibiting the occurrence of inflammation and preventing the spread of inflammatory reaction.

Recent studies have demonstrated that JNK is a crucial signaling pathway that mediates the cellular stress response, taking part in both cell proliferation and apoptosis. After JNK phosphorylation, it can regulate Bcl-2 family proteins, upregulate antiapoptotic protein Bcl-2, downregulate proapoptotic protein Bax, activate Caspase-8 and Caspase-9, and subsequently activate Caspase-3 [117,118]. Caspase-3 is the predominant shearing enzyme in the process of apoptosis, which cleaves cell proteins, destroys the cytoskeleton, and cleaves nuclear DNA [119]. *O. violaceus* can inhibit JNK phosphorylation, activate the AMPK/AKT/GSK-3β pathway, and inhibit Caspase-3 activation to form Cleaved Caspase-3, thus inhibiting apoptosis and protecting the liver from damage.

Research has indicated that radiation can indirectly impact water molecules within living organisms, resulting in their ionization and generating a significant quantity of free radicals. These free radicals interact with other substances in the body, leading to apoptosis and potentially contributing to the development of various harmful diseases [73]. Ferroptosis, as one of the programmed cell death modes, is mainly related to intracellular iron accumulation, glutathione (GSH) depletion, and lipid peroxidation [120], and studies [121] have shown that ferroptosis may become a potential target for inflammation-related diseases. Radiation generates a large amount of ROS to promote lipid peroxidation while inducing ACSL4 expression to increase the biosynthesis of PUFA-PL, thereby, in turn, inducing cell ferroptosis [122] (see Figure 13). It is speculated that the antiradiation and antiferroptosis effects of *O. violaceus* are also closely related to its ability to resist oxidative stress and inhibit inflammation. The Nrf2 signaling pathway not only plays a crucial role in maintaining redox homeostasis within the body but also serves as a key mediator in various essential metabolic pathways. These include lipid metabolism, iron/hemoglobin metabolism, and apoptosis [123]. Nrf2 also exerts a wide range of cytoprotective functions in anti-inflammatory, antitumor, antiapoptotic, and organ-protective activities [124,125]. It has been reported that Nrf2, a key mitigator of lipid peroxidation and ferroptosis, plays a protective role during ferroptosis in hepatocellular carcinoma cells by mediating the p62-Keap1-Nrf2 pathway and its elevated expression level can effectively protect against ferroptosis in hepatoma cells [126]. It has also been found that the NF-κB pathway is also involved in Erastin-induced ferroptosis in colon cancer cells and SPINK4 mediates ferroptosis by regulating the NF-κB pathway [127]. *O. violaceus* may resist oxidative and inflammatory damage and then exert antiradiation, antiferroptosis, and other pharmacological activities by mediating the Keap1-Nrf2/ARE signaling pathway as well as the NF-κB pathway, elevating Nrf2 expression, up-regulating HO-1 and GSH levels, and down-regulating TNF-α.

*O. violaceus* is a kind of medicinal resource plant with a high development value because of its rich nutritive value and health care function. It contains quercetin, catechin, and other antioxidant active ingredients. The alkaloids of *O. violaceus* have great benefits to human health such as anti-inflammatory, antioxidant, and antitumor, and have a protective effect on the liver and can be used to treat diseases. In particular, the new skeleton alkaloid orychophragine D shows significant antiradiation activity. Herbs are rarely used alone and are often combined with other herbs in Chinese medicine theory. Over recent years, there has been a growing trend of integrating herbs with conventional chemical drugs in clinical practice, resulting in significant therapeutic benefits. Thus, *O. violaceus* may have a superimposed effect when used in conjunction with conventional antiradiation drugs. Through the above findings, we can infer the mechanism of antiradiation and antiferroptosis, reveal the internal mechanism of its traditional curative effect by using clinical key indicators and data changes, and provide a scientific basis for disease treatment, which will help position *O. violaceus* as a promising candidate for future pharmaceutical development.

*O. violaceus* is known to be dense in fatty acids but has very little current application and potentially very high economic value. *O. violaceus* oil appears to have special lubricating properties due to the dihydroxylation of its major fatty acids and the presence of estolides in the oil. *O. violaceus* has the potential to be an industrial crop with oil properties distinct from those of castor oil as a chemical feedstock [128]. In addition, it is also rich in vitamins, amino acids, trace elements, etc., which can be used as a special dietary supplement to provide elements that cannot be synthesized by the human body.

Despite the research progress in chemical composition and pharmacological effects, there is also insufficient research information and a distinct lack of in-depth and systematic studies on *O. violaceus*. Focusing on the isolation and purification of its compounds as well as their pharmacological activity requires additional study. Further isolation and identification of chemical components can lay the foundation for their pharmacological activities and development applications. The extraction, preparation, and purification processes should adopt green chemistry methods to minimize the use of organic reagents and mitigate environmental pollution. Highly active compounds can be prepared by chemical total synthesis, laying the foundation for future industrial production of drugs. In addition, most of the identified compounds have not yet been validated in vivo, despite in vitro activity assessments to indicate that they have pharmacological effects. The in vivo pharmacodynamic evaluation and molecular biological mechanism research are essential to promote traditional and modern drug development. The specific action mechanism, signaling pathways, and targets of these compounds is also a pressing issue to be further explored by scholars nationwide and internationally. Alternatively, it is necessary to determine whether there are synergistic or antagonistic effects between these bioactive compounds [129]. The sources of *O. violaceus* vary slightly in different regions, but there is no uniform quality standard system yet. Therefore, a comprehensive quality standard system should be established to provide a basis for the clinical application and resource development of *O. violaceus*. There are still numerous promising opportunities and challenges in the research filed related to *O. violaceus* in future. Moderate techniques such as the high-efficiency separation technique, metabolomics, spectrum-effect relationship, network pharmacology, and molecular docking can be combined in the future to further clarify the pharmacological mechanism of action of *O. violaceus*, screen its active compounds, and identify its quality markers, thus providing a basis for the development of *O. violaceus* [130].

## Figures and Tables

**Figure 1 molecules-29-01314-f001:**
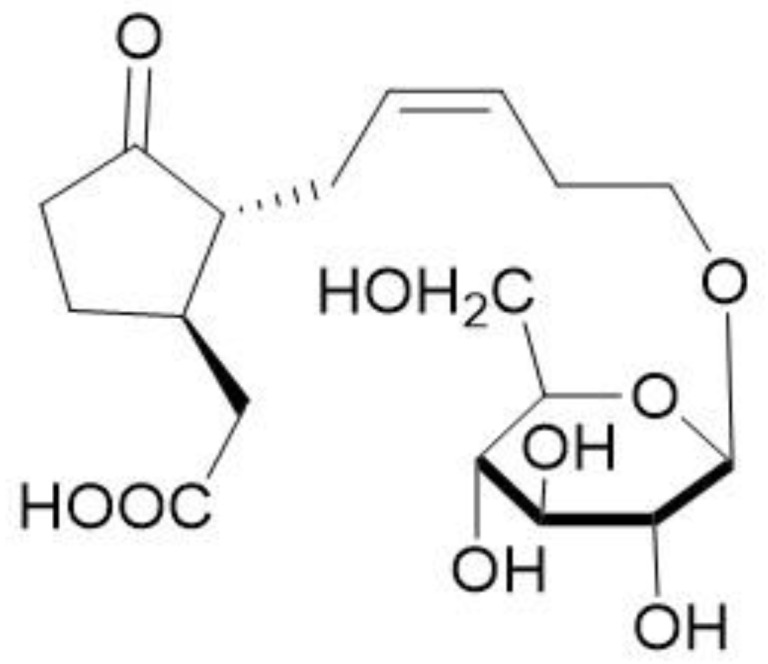
12-*O*-β-d-Glucopyranosyl-12-hydroxyjasmonic acid.

**Figure 2 molecules-29-01314-f002:**
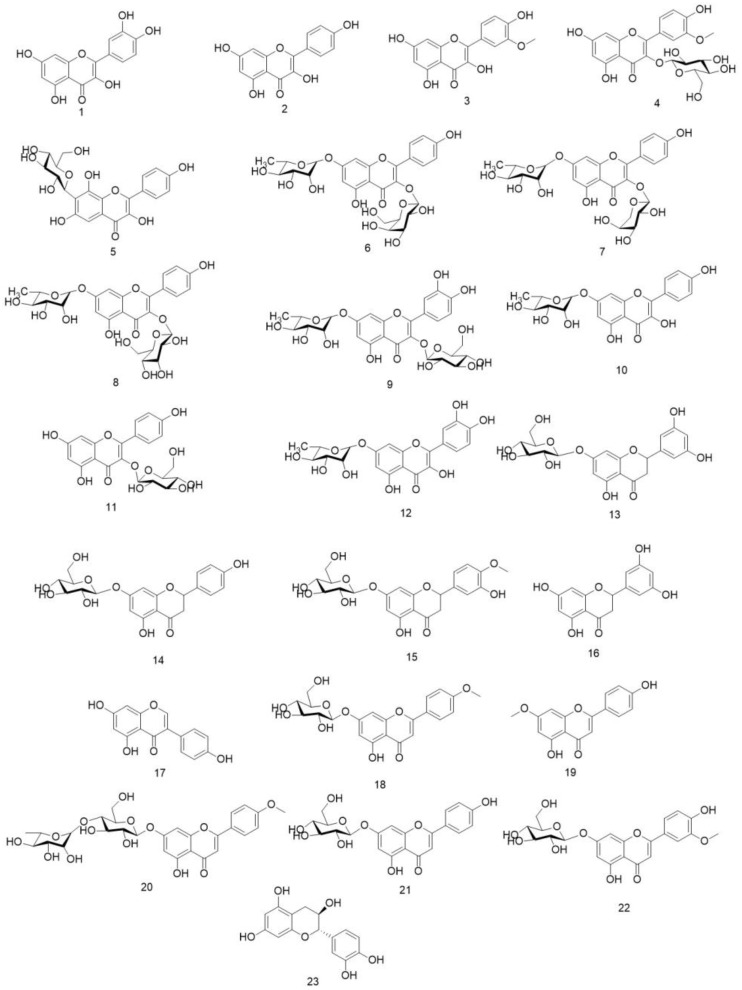
Structures of flavonoids in *O. violaceus*.

**Figure 3 molecules-29-01314-f003:**
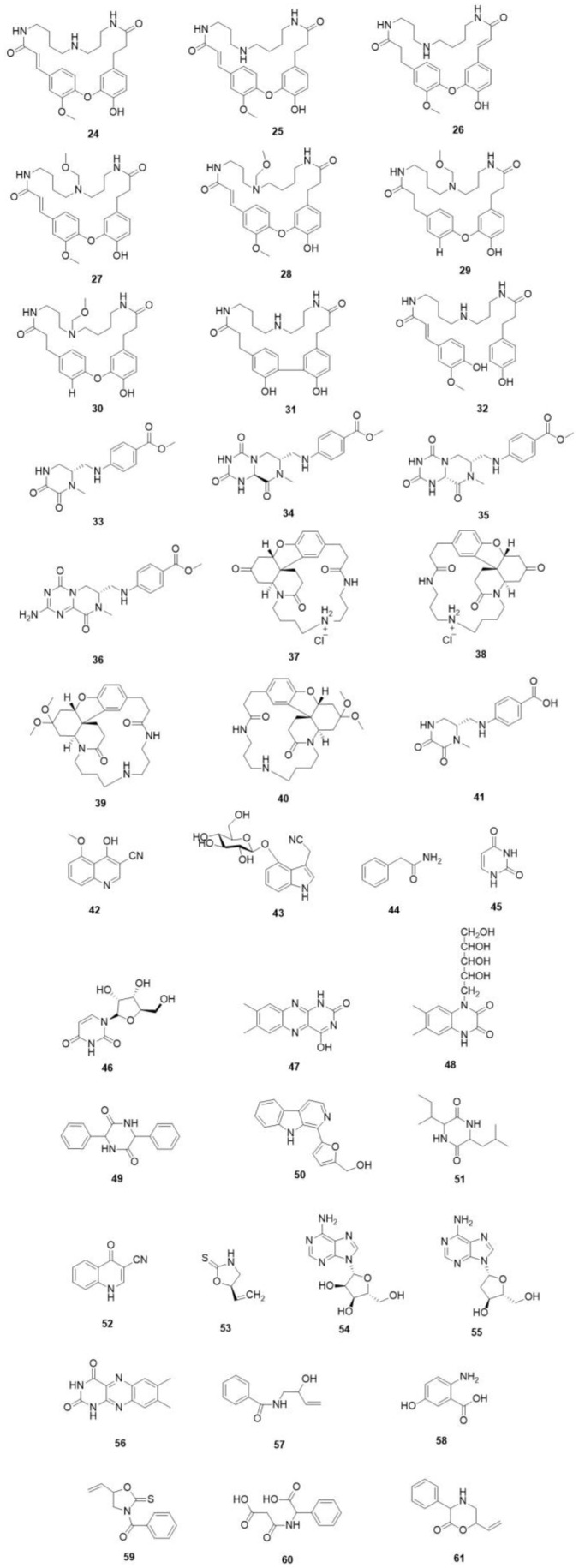
Structures of alkaloids in *O. violaceus*.

**Figure 4 molecules-29-01314-f004:**
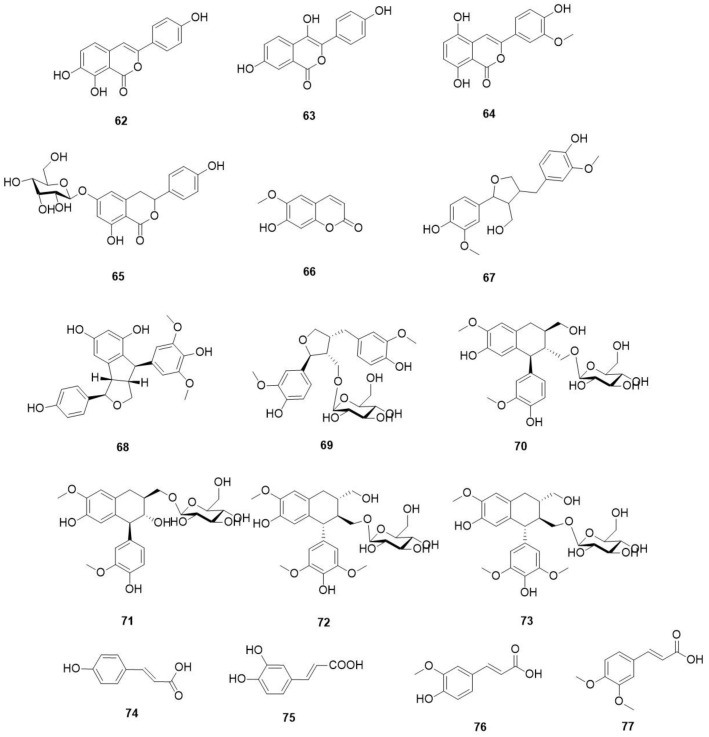
Structures of phenylpropanoids in *O. violaceus*.

**Figure 5 molecules-29-01314-f005:**
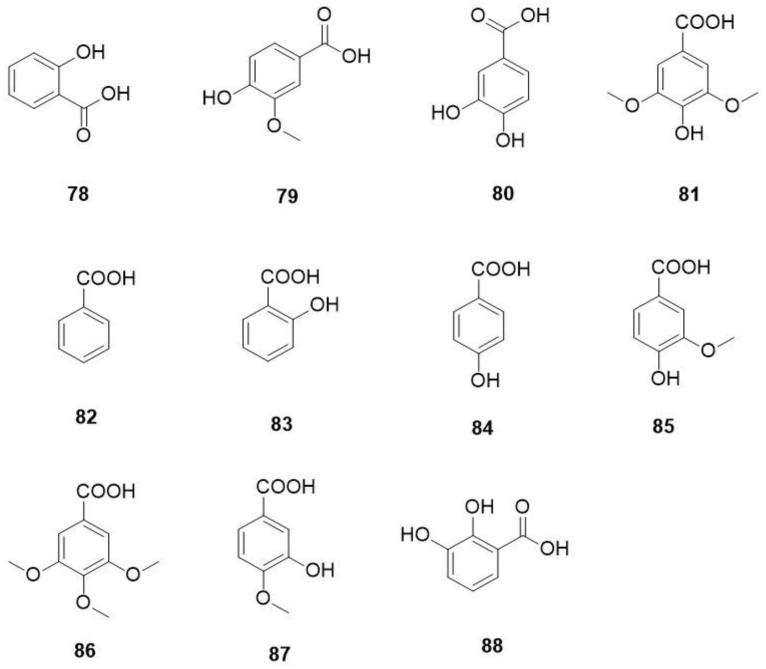
Structures of phenolic acids in *O. violaceus*.

**Figure 6 molecules-29-01314-f006:**
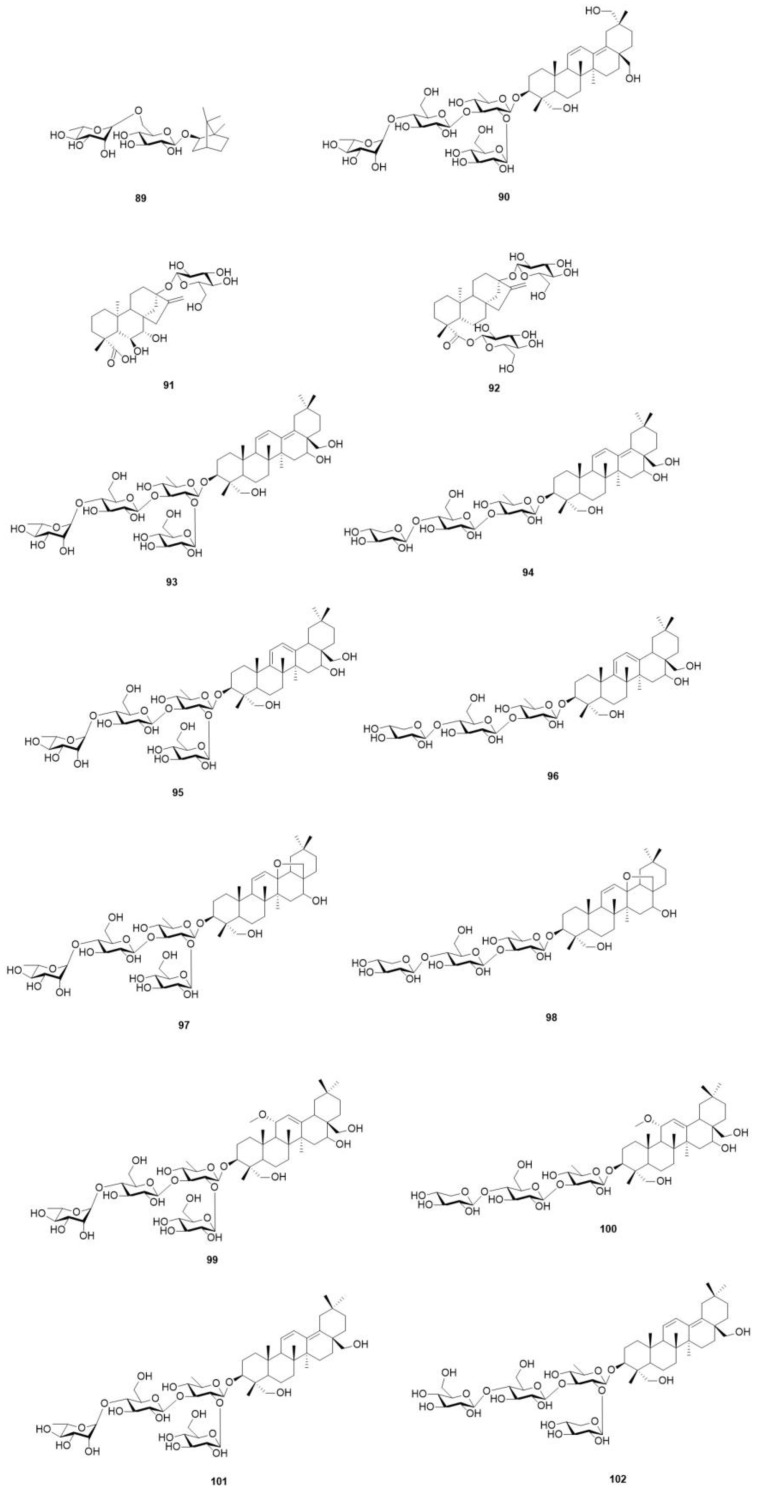
Structures of terpenoids in *O. violaceus*.

**Figure 7 molecules-29-01314-f007:**
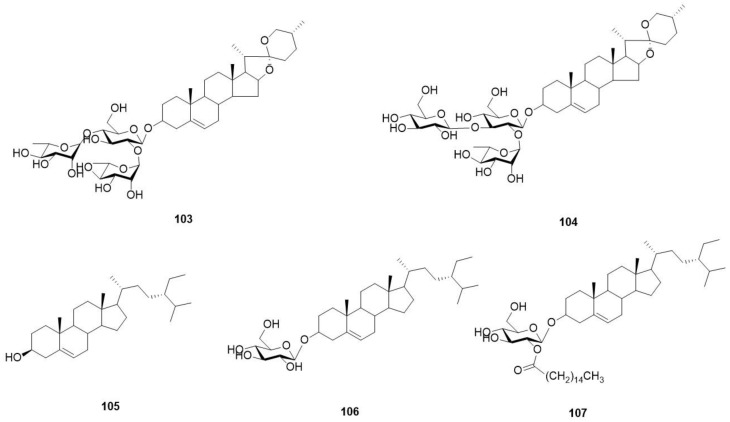
Structures of steroids in *O. violaceus*.

**Figure 8 molecules-29-01314-f008:**
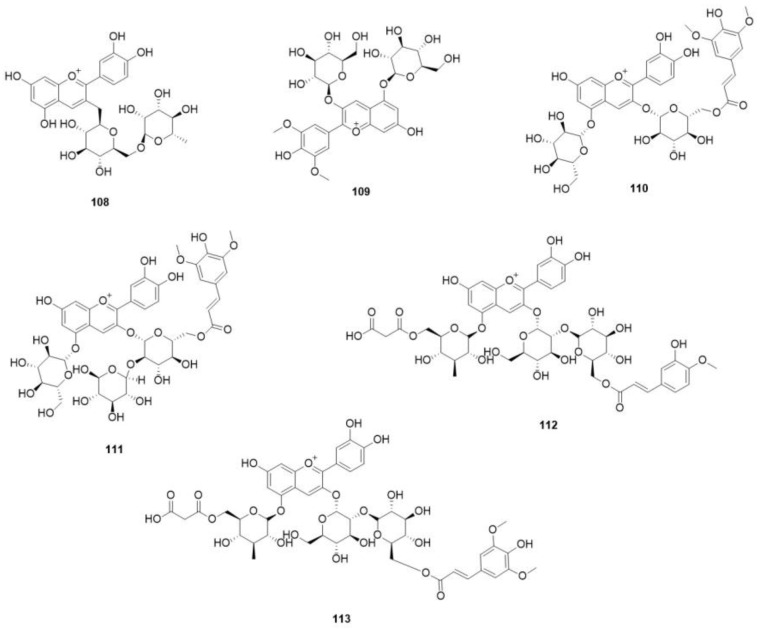
Structures of anthocyanins in *O. violaceus*.

**Figure 9 molecules-29-01314-f009:**
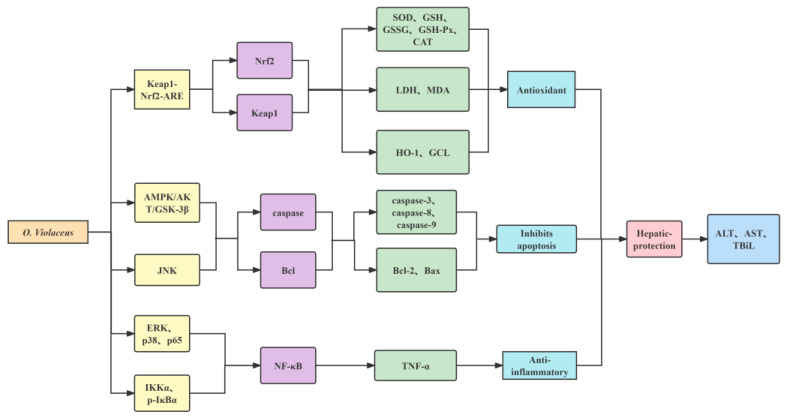
Mechanism of hepatic-protective action of *O. Violaceus*.

**Figure 10 molecules-29-01314-f010:**
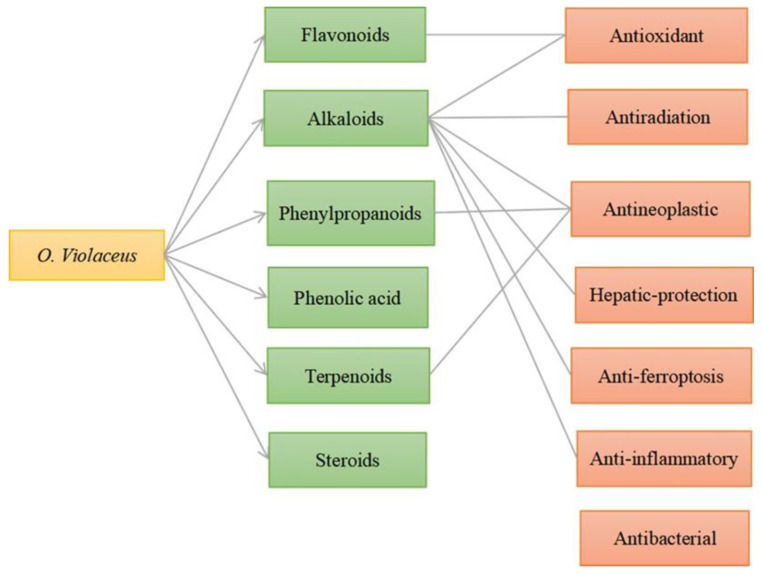
Correlation diagram of “chemical constituents-pharmacological activity” of *O. Violaceus*.

**Figure 11 molecules-29-01314-f011:**
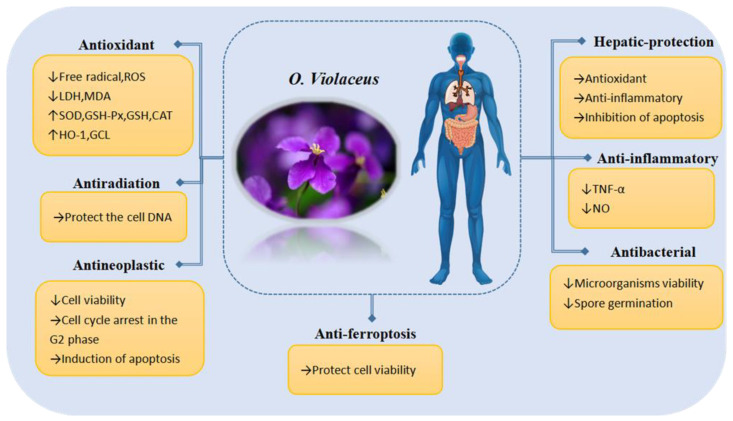
Overview of bioactivity and potential mechanism of action of *O. violaceus*. “↑”: up-regulation; “↓”: down-regulation.

**Figure 12 molecules-29-01314-f012:**
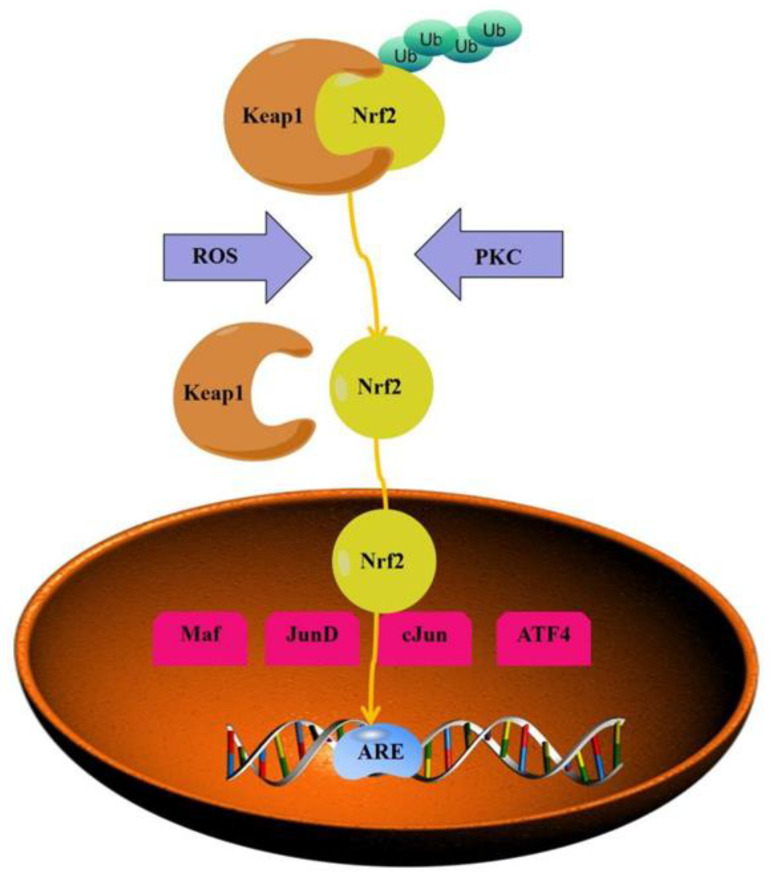
Nrf2 signaling pathway diagram.

**Figure 13 molecules-29-01314-f013:**
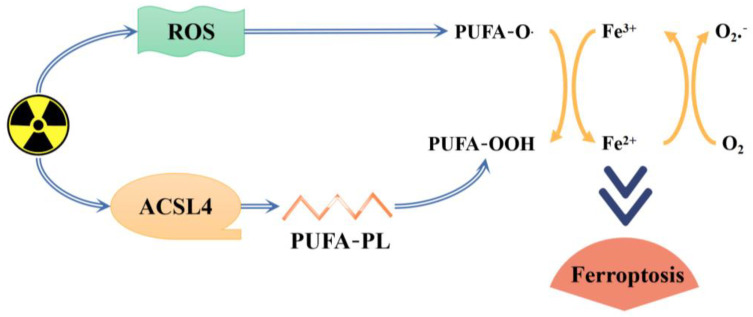
Diagram of radiation-induced ferroptosis.

**Table 1 molecules-29-01314-t001:** Comparison of the vitamin content of *O. violaceus* and *Brassica chinensis*.

Stage	Sample	V_C_ *	V_B1_ *	V_B2_ *	V_E_ *	V_K_ #	V_B6_ #	β-Carotene #
Seedling	*O. violaceus*	132.38	0.06	0.13	0.08	0.06	1.14	1.09
	*B.chinensis*	29.70	0.04	0.11	0.05	0.08	0.78	0.57
Moss	*O. violaceus*	101.62	0.05	0.11	0.11	0.06	0.84	1.31
	*B. chinensis*	44.57	0.04	0.09	0.10	0.07	0.67	1.01

*: fluorometric method; #: HPLC.

**Table 2 molecules-29-01314-t002:** Comparison of the main vitamin content of *O. violaceus* with other plants of Brassica.

Sample	β-Carotene ^@^	V_B1_ *	V_B2_ *	V_C_ *
*O. violaceus*	1620	0.10	0.05	27
*Brassica oleracea* var. *albiflora* Kuntze	3450	0.20	0.00	78
*Brassica campestris* var. *purpuraria*	1255	0.10	0.06	21
*Brassica caulorapa* Pasq.	20	0.04	0.82	41
Baby Bok Choy	1680	0.02	0.00	26
*Brassica oleracea* var. *botrytis* Linnaeus	20	0.08	0.08	81
*Brassica rapa* var. *glabra* Regel	40	0.08	0.04	28
*Brassica juncea* (L.) Czern. et Coss. Var. *foliosa* L. H. Bailey	1700	0.02	0.10	72
*Brassicajuncea* (L.) Czern. et Coss. Var. *multiceps* Tsen et Lee	310	0.03	0.11	31

^@^: paper chromatography; *: fluorometric method.

**Table 3 molecules-29-01314-t003:** Amino acid content (mg/g).

Amino Acid	Content	
	Seedling	Moss
* Ile	42.30	43.99
* Leu	79.27	84.03
* Phe	38.66	40.21
* Lys	58.49	61.99
* Thr	41.92	44.85
* Val	63.16	63.79
* Met	6.66	6.56
* Trp	13.54	12.32
Ser	42.18	43.02
Tyr	27.51	28.89
Ala	60.62	62.44
Glu	112.33	117.95
His	20.09	20.09
Arg	41.71	44.21
Pro	114.88	92.05
Cys	6.74	6.94
Asp	78.42	83.13
Gly	51.29	52.83

*: Essential amino-acid.

**Table 4 molecules-29-01314-t004:** Contents of elements in different organs of *O. violaceus* (μg/g).

Elements		Flower	Root	Leaf
Macroelement	B	19.69	12.01	22.05
	Na	210.53	199.25	240.61
	Mg	1204.99	380.79	1271.30
	Si	322.58	239.72	207.08
	P	3864.03	1747.57	2613.93
	K	12,911.21	8075.82	10,043.72
	Ca	4750.79	3121.87	12,834.32
Secondary element	Mn	32.64	16.31	86.86
	Fe	234.52	123.06	1776.91
	Cu	8.56	3.34	4.98
	Zn	44.10	16.65	28.75
	Br	2.69	2.02	7.86
	Sr	23.13	35.13	88.83
	I	0.95	1.90	1.30
Heavy metal element	Al	170.03	131.51	95.07
	Ti	15.38	12.30	43.33
	Cr	2.14	2.22	221.68
	Ni	1.66	0.90	18.70
	Pb	1.357	0.184	0.679
Trace beneficial element	Li	0.301	0.198	0.414
	Co	0.374	0.174	0.578
	Mo	0.885	0.576	1.476
Rare earth element	La	0.248	0.135	0.114
	Ce	0.717	0.420	0.343
	Nd	0.185	0.102	0.087
	Sc	0..233	0.158	0.179

**Table 5 molecules-29-01314-t005:** Fatty acid composition of *O. violaceus* seed oil.

Compounds	Molecular Formula	Relative Area Percentage (%)	
		MIBSE	SE
Palmitic acid	C_16_H_32_O_2_(C_16:0_)	8.84	9.11
Linoleic acid	C_18_H_32_O_2_(C_18:2_)	55.62	54.93
Oleic acid	C_18_H_34_O_2_(C_18:1_)	15.85	15.58
Stearic acid	C_18_H_36_O_2_(C_18:0_)	3.67	4.19
Linolenic acid	C_18_H_30_O_2_(C_18:3_)	4.02	3.78
cis-11,14-Eicosenoic acid	C_20_H_36_O_2_(C_20:2_)	0.51	0.56
11-Eicosenoic acid	C_20_H_38_O_2_(C_20:1_)	3.87	4.11
Arachidic acid	C_20_H_40_O_2_(C_20:0_)	2.94	3.11
Behenic acid	C_22_H_44_O_2_(C_22:0_)	1.18	1.31
Nervonic acid	C_24_H_46_O_2_(C_24:1_)	2.40	2.32
Tetracosenoic acid	C_24_H_48_O_2_(C_24:0_)	1.10	1.01

**Table 6 molecules-29-01314-t006:** Comparison of quality of *O. violaceus* and rape.

Plant	Oil Content (%)	Crude Protein Content (%)	Glucosinolate Content (μg/g)
*O. violaceus*	50.29	52.32	35.24
Ning-You-7-Hao	41.71	34.20	69.67
Westar	42.90	37.58	6.77

**Table 7 molecules-29-01314-t007:** Flavonoids of *O. violaceus*.

No.	Compound	Type	Formula	Mol. wt.	References
1	Quercetin	flavonols	C_15_H_10_O_7_	302.02	[37]
2	Kaempferol	flavonols	C_15_H_10_O_6_	286.05	[37]
3	Isorhamnetin	flavonols	C_16_H_12_O_7_	316.06	[37]
4	Isorhamnetin-3-*O*-β-d-glucopyranoside	flavonols	C_22_H_22_O_12_	478.11	[41]
5	3,4′,6,8-tetyahydroxy-flavone-7-C-glucoside	flavonols	C_21_H_20_O_11_	448.10	[42]
6	kaeMpferol 3-*O*-β-d-galactosyl-7-*O*-α-L-rhamnoside	flavonols	C_27_H_30_O_15_	594.16	[42]
7	kaeMpferol 3-*O*-β-d-arabinosyl-7-*O*-α-L-rhamnoside	flavonols	C_26_H_28_O_14_	564.15	[42]
8	kaeMpferol 3-*O*-β-d-glucopyranosyl-7-*O*-α-L-rhamnoside	flavonols	C_27_H_30_O_15_	594.16	[42]
9	Quercetin 3-*O*-β-d-glucopyranosyl-7-*O*-α-L-rhamnoside	flavonols	C_27_H_30_O_16_	610.15	[42]
10	Kaempferol-7-*O*-α-L-rhamnoside	flavonols	C_21_H_20_O_10_	432.11	[42]
11	kaempferol-3-*O*-β-d-glucoside	flavonols	C_21_H_20_O_11_	448.10	[42]
12	Quercetin-7-*O*-α-L-rhamnoside	flavonols	C_21_H_20_O_11_	448.10	[42]
13	5,7,3′,5′-tetyahydroxy-flavone-7-*O*-β-d-glucoside	flavanones	C_21_H_22_O_11_	450.12	[41]
14	aromadendrin-7-*O*-β-d-glucopyranoside	flavanones	C_21_H_22_O_10_	434.12	[41]
15	hesperetin-7-*O*-β-d-glucopyranoside	flavanones	C_22_H_24_O_11_	464.12	[41]
16	3′,5′,5,7-tetyahydroxy-dihydroflavone	flavanones	C_15_H_12_O_6_	288.06	[9]
17	Geristein	isoflavones	C_15_H_10_O_5_	270.05	[11]
18	acactin-7-*O*-β-d-glucoside	flavones	C_22_H_22_O_10_	446.12	[11]
19	Genkwanin	flavones	C_16_H_12_O_5_	284.07	[41]
20	acacetin-7-*O*-neohesperidoside	flavones	C_28_H_32_O_14_	592.18	[41]
21	apigenin-7-*O*-β-d-glucoside	flavones	C_21_H_20_O_10_	432.11	[41]
22	chrysoeriol-7-*O*-β-d-glucoside	flavones	C_22_H_22_O_11_	462.12	[41]
23	Catechin	flavan-3-ols	C_15_H_12_O_6_	288.06	[43]

**Table 8 molecules-29-01314-t008:** Alkaloids of *O. violaceus*.

No.	Compound	Formula	Mol. wt.	References
24	orychophragmuspine A	C_26_H_33_N_3_O_5_	467.27	[12]
25	orychophragmuspine B	C_26_H_33_N_3_O_5_	467.24	[12]
26	orychophragmuspine C	C_26_H_33_N_3_O_5_	467.24	[12]
27	orychophragmuspine D	C_28_H_35_N_3_O_6_	509.25	[12]
28	orychophragmuspine E	C_28_H_35_N_3_O_6_	509.25	[12]
29	orychophragmuspine F	C_27_H_35_N_3_O_5_	481.26	[12]
30	orychophragmuspine G	C_27_H_35_N_3_O_5_	481.26	[12]
31	orychophragmuspine H	C_25_H_33_N_3_O_4_	439.25	[12]
32	orychophragmuspine I	C_26_H_35_N_3_O_5_	469.26	[46]
33	orychophragine A	C_14_H_17_N_3_O_4_	291.12	[47]
34	orychophragine B	C_16_H_19_N_5_O_5_	361.14	[47]
35	orychophragine C	C_16_H_19_N_5_O_5_	361.14	[47]
36	orychophragine D	C_16_H_18_N_6_O_4_	358.14	[48]
37	(+)-orychoviolines A	C_25_H_33_O_4_N_3_	439.25	[50]
38	(−)-orychoviolines A	C_25_H_33_O_4_N_3_	439.25	[50]
39	(+)-orychoviolines B	C_27_H_39_O_5_N_3_	485.29	[50]
40	(−)-orychoviolines B	C_27_H_39_O_5_N_3_	485.29	[50]
41	demethylorychophragine A	C_13_H_15_N_3_O_4_	277.11	[51]
42	4-hydroxy-5-methoxy-3-quinolinecarbonitrile	C_11_H_8_N_2_O_2_	200.06	[42]
43	Cappariloside A	C_16_H_18_N_2_O_6_	334.12	[42]
44	2-Phenylacetamide	C_8_H_9_NO	135.07	[42]
45	Uracil	C_4_H_4_N_2_O_2_	112.03	[42]
46	Uridine	C_9_H_12_N_2_O_6_	244.07	[42]
47	isolumichrome	C_12_H_10_N_4_O_2_	242.08	[42]
48	1-ribosyl-2-3-diketo-1,2,3,4-tetrahydro-6,7-dimethyl-quinoxaline	C_15_H_20_N_2_O_6_	324.13	[42]
49	3,6-dibenzyl-2,5-dioxopiperazine	C_16_H_14_N_2_O_2_	266.11	[42]
50	perlolyrine	C_16_H_12_N_2_O_2_	264.09	[42]
51	DL-isoleucyl-leucyl anhydride	C_12_H_22_N_2_O_2_	226.17	[42]
52	4-oxo-1,4-dihydroquinoline-3-carbonitrile	C_10_H_6_N_2_O	170.05	[42]
53	epigoitrin	C_5_H_7_NOS	129.02	[50]
54	Adenosine	C_10_H_13_N_5_O_4_	267.10	[9]
55	2′-Dideoxyadenosine	C_10_H_13_N_5_O_3_	251.10	[9]
56	7,8-Dimethylalloxazine	C_12_H_10_N_4_O_2_	242.08	[52]
57	N-2-hydroxy-3-butenyl-Benzamide	C_11_H_13_NO_2_	191.09	[53]
58	2-Amino-5-hydroxybenzoic acid	C_7_H_7_NO_3_	153.04	[53]
59	N-benzoyl-epigoitri	C_12_H_11_NO_2_S	233.05	[53]
60	α-[(2-carboxyacetyl)amino]-Benzeneacetic acid	C_11_H_11_NO_5_	237.06	[53]
61	3-phenyl-6-vinylmorpholin-2-one	C_12_H_13_NO_2_	203.09	[53]

**Table 9 molecules-29-01314-t009:** Phenylpropanoids of *O. Violaceus*.

No.	Compound	Type	Formula	Mol. wt.	References
62	orychophramarin A	coumarin	C_15_H_10_O_5_	270.05	[13]
63	orychophramarin B	coumarin	C_15_H_10_O_5_	270.05	[13]
64	orychophramarin C	coumarin	C_16_H_12_O_6_	300.06	[13]
65	orychophramarin D	coumarin	C_21_H_22_O_10_	434.12	[13]
66	7-hydroxy-6-methoxy coumarin	coumarin	C_10_H_8_O_4_	192.04	[13]
67	lariciresinol	lignans	C_20_H_24_O_6_	360.16	[13]
68	11-deoxykompasinol A	lignans	C_25_H_24_O_7_	436.15	[31]
69	(+)-lariciresinol-9-*O*-β-d-glucopyranoside	lignans	C_26_H_34_O_11_	522	[31]
70	(+)-isolariciresinol-9′-*O*-β-d-glucopyranoside	lignans	C_26_H_34_O_11_	522	[31]
71	(+)-isolariciresinol-9-*O*-β-d-glucopyranoside	lignans	C_26_H_34_O_11_	522	[31]
72	(−)-isolariciresinol-9′-*O*-β-d-glucopyranoside	lignans	C_26_H_34_O_11_	522	[31]
73	(−)-5′-methoxyisolariciresinol-9′-*O*-β-d-glucopyranoside	lignans	C_27_H_36_O_11_	434.16	[31]
74	p-hudroxycoumaric	phenylpropanoic acids	C_9_H_8_O_3_	164.05	[42]
75	caffeic acid	phenylpropanoic acids	C_9_H_8_O_4_	180.04	[14]
76	trans-Ferulic acid	phenylpropanoic acids	C_10_H_10_O_4_	194.06	[42]
77	3,4-dimethoxy cinnamic acid	phenylpropanoic acids	C_11_H_12_O_4_	208.07	[42]

**Table 10 molecules-29-01314-t010:** Phenolic acids of *O. Violaceus*.

No.	Compound	Type	Formula	Mol. wt.	References
78	salicylic acid	benzoic acid	C_7_H_6_O_3_	138.03	[14]
79	vanillic acid	benzoic acid	C_8_H_8_O_4_	168.04	[14]
80	protocatechuic acid	benzoic acid	C_7_H_6_O_4_	154.03	[14]
81	syringate	benzoic acid	C_9_H_10_O_5_	198.05	[14]
82	benzonic acid	benzoic acid	C_7_H_6_O_2_	122.04	[41]
83	2-hydroxybenzoic acid	benzoic acid	C_7_H_6_O_3_	138.03	[41]
84	4-hydroxybenzoic acid	benzoic acid	C_7_H_6_O_3_	138.03	[41]
85	4-hydroxy-3-methoxy-benzoic acid	benzoic acid	C_8_H_8_O_4_	168.04	[41]
86	Eudesmic acid	benzoic acid	C_10_H_12_O_5_	212.07	[41]
87	Isovanillic acid	benzoic acid	C_8_H_8_O_4_	168.04	[42]
88	2,3-dihydroxybenzoic acid	benzoic acid	C_7_H_6_O_4_	154.03	[42]

**Table 11 molecules-29-01314-t011:** Terpenoids of *O. Violaceus*.

No.	Compound	Type	Formula	Mol. wt.	References
89	orychovioside A	monoterpene glycoside	C_22_H_38_O_10_	462.25	[46]
90	orychoside A	triterpene saponin	C_54_H_88_O_22_	1088.58	[15]
91	orychoside B	Diterpene glycosides	C_26_H_40_O_10_	512.26	[15]
92	rubusoside	Diterpene glycosides	C_32_H_50_O_13_	642.33	[15]
93	mimengoside C	triterpene saponin	C_54_H_88_O_22_	1088.58	[15]
94	mimengoside D	triterpene saponin	C_47_H_76_O_17_	912.51	[15]
95	mimengoside E	triterpene saponin	C_54_H_88_O_22_	1088.58	[15]
96	mimengoside F	triterpene saponin	C_47_H_76_O_17_	912.51	[15]
97	buddlejasaponin I	triterpene saponin	C_54_H_88_O_22_	1088.58	[15]
98	buddlejasaponin III	triterpene saponin	C_47_H_76_O_17_	912.51	[15]
99	buddlejasaponin Ia	triterpene saponin	C_55_H_92_O_23_	1120.60	[15]
100	buddlejasaponin IIIa	triterpene saponin	C_48_H_80_O_18_	944.53	[15]
101	3-*O*-α-L-rhamnose-(1→4)-β-d-glucose(1→3)-[β-d-glucose-(1→2)]-β-d-fucose-23,28-dihydroxysaikosaponin-11,13(18)-diene	triterpene saponin	C_54_H_88_O_21_	1072.58	[15]
102	3β-*O*-(β-d-glucose-(1→4)-β-d-glucose(1→3)-[β-d-glucose-(1→2)]-β-d-fucose-23,28-dihydroxysaikosaponin-11,13(18)-diene	triterpene saponin	C_53_H_86_O_21_	1058.57	[15]

**Table 12 molecules-29-01314-t012:** Steroids of *O. Violaceus*.

No.	Compound	Formula	Mol. wt.	References
103	dioscin	C_45_H_72_O_16_	868.48	[11]
104	gracillin	C_45_H_72_O_17_	884.48	[11]
105	β-Sitosterol	C_29_H_50_O	414.39	[41]
106	Daucosterol	C_35_H_60_O_6_	576.44	[41]
107	β-Sitosterol-3-*O*-β-d-glucopyranoside-20-*O*-palmitate	C_51_H_90_O_7_	814.67	[41]

**Table 13 molecules-29-01314-t013:** Anthocyanins of *O. Violaceus*.

No.	Compound	Formula	Mol. wt.	References
108	Centaureidin 3-rutinoside	C_28_H_33_O_14_^+^	593.19	[67]
109	Malvidin 3,5-diglucoside	C_29_H_35_O_17_^+^	655.19	[67]
110	Centaureidin 3-coumaroylglucoside-5-glucoside	C_38_H_41_O_20_^+^	817.22	[67]
111	Centaureidin 3-p-coumaroyl diglucoside-5-glucoside	C_43_H_49_O_25_^+^	965.26	[67]
112	Centaureidin 3-feruloylsophoroside-5-malonyl glucoside	C_47_H_53_O_26_^+^	1033.28	[67]
113	Centaureidin 3-coumaroyl locust glycoside-5-malonyl glucoside	C_48_H_55_O_27_^+^	1063.29	[67]

**Table 14 molecules-29-01314-t014:** Antitumor effect of *O. Violaceus*.

Compounds	IC_50_ (μM)					References
	Hela	HepG2	A549	HCT-116	SH-SY5Y	
11-deoxykompasinol A	9.43 ± 0.62	18.26 ± 1.31	/	/	/	[31]
(+)-lariciresinol-9-*O*-β-d-glucopyranoside	17.82 ± 1.17	36.78 ± 1.21	/	/	/	[31]
β-d-glucopyranosyl-12-hydroxyjasmonate	22.20 ± 2.24	29.10 ± 2.45	/	/	/	[31]
orychophragine A	11.91 ± 0.46	7.73 ± 0.55	10.79 ± 0.86	9.93 ± 0.71	/	[47]
orychoside A	/	7.13 ± 0.10	8.77 ± 0.31	/	/	[41]
mimengoside C	/	7.82 ± 0.17	6.78 ± 0.21	/	/	[41]
mimengoside D	/	9.62 ± 0.68	11.63 ± 0.50	/	/	[41]
mimengoside E	/	16.82 ± 1.46	14.03 ± 0.49	/	/	[41]
mimengoside F	/	22.20 ± 1.24	19.10 ± 0.45	/	/	[41]
buddlejasaponin I	/	0.78 ± 0.21	0.48 ± 0.04	/	/	[41]
buddlejasaponin III	/	1.06 ± 0.09	0.96 ± 0.12	/	/	[41]
orychophramarin A	8.91 ± 0.65	/	/	5.10 ± 0.46	/	[13]
orychophramarin B	23.47 ± 1.37	/	/	28.30 ± 0.91	/	[13]
orychophramarin C	14.24 ± 0.67	/	/	10.12 ± 0.54	/	[13]
orychophramarin D	>50	/	/	44.71 ± 1.93	/	[13]
perlolyrine	51.3 ± 1.32	/	/	/	23.4 ± 2.88	[42]

## Data Availability

Not applicable.

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
