# Peer review of "Recent Advances in the Nutritional Value, Chemical Compositions, Pharmacological Activity, and Application Value of Orychophragmus violaceus: A Comprehensive Review"

_molecules, 2024, doi:10.3390/molecules29061314_

Round 1

Reviewer 1 Report

Comments and Suggestions for Authors

The manuscript presents an updated review of compounds isolated from Orychophragmus violaceus aimed to summarize recent research progress dealing in particular with compounds of nutritional and biological value.

The manuscript does not describe new research activity, its aim is to provide an updated literature review on recent scientific publications dealing with natural compounds of nutritional and biological value isolated from Orychophragmus violaceus.
As such, the manuscript does not contain original research, but it represents a consistent and well-organized reference paper for scientists involved in further research in this area.
I didn’t find a previous comprehensive and recent review on the same subject area; thus, this manuscript represents a useful addition.
The literature search is based on consultation of several scientific data banks, and it appears satisfactory to the scope of the manuscript. References are consistent and appropriate. Figures and tables provide a rational and rapid identification of isolated compounds and of their literature references. The text summarizes adequately recent research progresses related to compounds with nutritional and biological value.

I just have a few suggestions to deal with:

Page 5, Figure 1: I suggest using as name 12-O-β-D-Glucopyranosyl-12-hydroxyjasmonic acid.

Page 6: In some cases, O. violaceus needs to be put in Italic text.

Page 12, Table 9: I suggest using phenylpropanoic acids instead of phenylpropionic acids.

Comments on the Quality of English Language

The manuscript is clear and readable. I am not a native English speaker, so I do not feel qualified for an accurate assessment of the quality of English. I think a minor editing of English language might be required.

Author Response

Thanks very much for taking your time to review this manuscript. All your suggestions are of great significance to my thesis writing and scientific research work. We have studied comments carefully and have made corrections which we hope meet with approval. Revised portions are highlighted in the manuscript.

Point 1: Page 5, Figure 1: I suggest using as name 12-O-β-D-Glucopyranosyl-12-hydroxyjasmonic acid.

Response 1:

Original Figure 1 of page 5: "Structures of β-D-glucopyranosyl-12-hydroxyjasmonate" has been changed to "Structures of 12-O-β-D-Glucopyranosyl-12-hydroxyjasmonic acid". The revised places have been highlighted. Please see in line 188 of page 6 in the revised manuscript.

Point 2: Page 6: In some cases, O. violaceus needs to be put in Italic text.

Response 2:

Original line 176, 194, 199, 203, 205, 209, 276, Table 5: "O. violaceus" has been changed to "O. violaceus". The revised places have been highlighted.

Point 3: Page 12, Table 9: I suggest using phenylpropanoic acids instead of phenylpropionic acids

Response 3:

Original Page 12, Table 9: "phenylpropionic acids" has been changed to "phenylpropanoic acids". The revised places have been highlighted. Please see in Page 11, Table 9 in the revised manuscript.

Once again, thank you for your sincere suggestion. Heartily, I am looking forward to receiving your affirmative reply.

Reviewer 2 Report

Comments and Suggestions for Authors

The authors attempt to summarize the nutritional value, chemical compositions, pharmacological activity, and application value of Orychophragmus violaceus from China and international literature. Overall, the review ideas appear justified. The manuscript is also well organized. I find the paper quite good. The listed are some comments regarding the submitted manuscript.

1.       In the introduction part: the nutritional value and application value of Orychophragmus violaceus should be provided more detail in the introduction part. It is difficult to find these point in the introduction section.

2.       Line 58-61:” Based on domestic and foreign research, flavonoids, alkaloids, phenylpropanoids, phenolic acids, and terpenoids have been identified as the main components of O. violaceus. → Please add the references. What is demonstrated that the main components of O. violaceus are flavonoids, alkaloids, phenylpropanoids, phenolic acids, and terpenoids?

3.       Line 75 - 76, Research methodology: What’s ACS and CNKI?

4.       Line 76: “We searched for information in both Chinese and English”. However, I could not found any reference by Chinese languages, please provide the databases as Chinese languages?

5.       Line 97-98: The legends of Table 1 and Table 2 should mentioned the method for analyzing β-carotene (HPLC or paper chromatography analysis).

6.       Line 174: What’s Glc in the Figure 1?

7.       Line 176, Table 5; Line 194; Line 199; 203; 205; 209; 276: O. violaceus → O. violaceus

8.       Line 256: Angelica Sinensis and Salvia miltiorrhiza → Angelica Sinensis and Salvia miltiorrhiza

9.       Line 329: Please provide the explanation about CAT, SOD, and GSH-Px

10.   Line 560, Figure 9: What’s the meaning of color in Figure 9?

Author Response

Thanks very much for taking your time to review this manuscript. All your suggestions are of great significance to my thesis writing and scientific research work. We have studied comments carefully and have made corrections which we hope meet with approval. Revised portions are highlighted in the manuscript.

Point 1: In the introduction part: the nutritional value and application value of Orychophragmus violaceus should be provided more detail in the introduction part. It is difficult to find these points in the introduction section.

Response 1:

It's in the revised manuscript. We have added a more detailed introduction to the nutritional value and application value of Orychophragmus violaceus in page 1-2, line 44-46 :"O. violaceus can be found everywhere along the roadsides in China and is commonly used as ornamental plant to both prevent soil erosion and beautify the environment." and line 70-73:"O. violaceus is rich in vitamin C, carotene and so on, which has potential edible and medicinal value for human body to prevent cold, cancer, cardiovascular and cerebrovascular diseases.". The revised places have been highlighted.

Point 2:  Line 58-61:” Based on domestic and foreign research, flavonoids, alkaloids, phenylpropanoids, phenolic acids, and terpenoids have been identified as the main components of O. violaceus. → Please add the references. What is demonstrated that the main components of O. violaceus are flavonoids, alkaloids, phenylpropanoids, phenolic acids, and terpenoids?

Response 2:

We have added the references in the revised manuscript: "Based on domestic and foreign research, flavonoids[10], alkaloids[12], phenylpropanoids[13], phenolic acids[14], and terpenoids[15] have been identified as the main components of O. violaceus.". The revised places have been highlighted.

Point 3: Line 75 - 76, Research methodology: What’s ACS and CNKI?

Response 3:

We are very sorry for not providing the full title. We have added these in the revised manuscript: " Journal of the American Chemical Society (ACS), China National Knowledge Infrastructure (CNKI)"The revised places have been highlighted. Please see in Page 2, line 80-82 in the revised manuscript.

Point 4: Line 76: “We searched for information in both Chinese and English”. However, I could not found any reference by Chinese languages, please provide the databases as Chinese languages?

Response 4:

China National Knowledge Infrastructure (CNKI) is the databases as Chinese languages. We translate relevant information from Chinese references into English for citations, for example, references [1],[2],[3], etc. in the revised manuscript.

Point 5: Line 97-98: The legends of Table 1 and Table 2 should mentioned the method for analyzing β-carotene (HPLC or paper chromatography analysis).

Response 5:

We have added the legends in page 3 Table 1 and Table 2 in the revised manuscript. The revised places have been highlighted.

Table 1. Comparison of the vitamin content of O. violaceus and Brassica chinensis.

Stage

Sample

VC*

VB1*

VB2*

VE*

VK#

VB6#

β-carotene#

seedling

O. violaceus

132.38

0.06

0.13

0.08

0.06

1.14

1.09

B.chinensis

29.70

0.04

0.11

0.05

0.08

0.78

0.57

moss

O. violaceus

101.62

0.05

0.11

0.11

0.06

0.84

1.31

B.chinensis

44.57

0.04

0.09

0.10

0.07

0.67

1.01

*:fluorometric method

#:HPLC

Table 2. Comparison of the main vitamin content of O. violaceus with other plants of Brassica.

Sample

β-carotene@

VB1*

VB2*

VC*

O. violaceus

1620

0.10

0.05

27

Brassica oleracea var. albiflora Kuntze

3450

0.20

0.00

78

Brassica campestris var. purpuraria

1255

0.10

0.06

21

Brassica caulorapa Pasq.

20

0.04

0.82

41

Baby Bok Choy

1680

0.02

0.00

26

Brassica oleracea var. botrytis Linnaeus

20

0.08

0.08

81

Brassica rapa var. glabra Regel

40

0.08

0.04

28

Brassica juncea (L.) Czern. et Coss. var. foliosa L. H. Bailey

1700

0.02

0.10

72

Brassicajuncea(L.) Czern. et Coss. var.multicepsTsen et Lee

310

0.03

0.11

31

@:paper chromatography

*:fluorometric method

Point 6: Line 174: What’s Glc in the Figure 1?

Response 6:

We have modified Figure 1 in the revised manuscript.

Point 7: Line 176, Table 5; Line 194; Line 199; 203; 205; 209; 276: O. violaceus → O. violaceus

Response 7:

Original line 176, 194, 199, 203, 205, 209, 276, Table 5: "O. violaceus" has been changed to "O. violaceus". The revised places have been highlighted.

Point 8: Line 256: Angelica Sinensis and Salvia miltiorrhiza → Angelica Sinensis and Salvia miltiorrhiza

Response 8:

Original line 256: " Angelica Sinensis and Salvia miltiorrhiza " has been changed to " Angelica Sinensis and Salvia miltiorrhiza ". The revised places have been highlighted. Please see in line 265 of page 12 in the revised manuscript.

Point 9: Line 329: Please provide the explanation about CAT, SOD, and GSH-Px

Response 9:

We are very sorry for not providing the full title. We have added these in the revised manuscript: " Catalase (CAT) , superoxide dismutase (SOD), and Glutathione peroxidase (GSH-Px)". The revised places have been highlighted. Please see in Page 16, line 336-337 in the revised manuscript.

Point 10: Line 560, Figure 9: What’s the meaning of color in Figure 9?

Response 10:

The colors in Figure 9 have no specific meaning, only for distinction and aesthetics.

Once again, thank you for your sincere suggestion. Heartily, I am looking forward to receiving your affirmative reply.

Reviewer 3 Report

Comments and Suggestions for Authors

The theme of the article is highly relevant in the context of research on plants with potential medicinal, nutritional, and diversified applications. The exploration of Orychophragmus violaceus covers important areas such as pharmacology, nutrition, and industrial applications (such as oil), which is significant for both science and industry.

The text presents good grammatical quality, with the appropriate use of scientific terms that contribute to its credibility and technical accuracy. However, there are some minor issues that could be revised to improve fluency and clarity:

1.      The presence of hyphens in words that would not normally be hyphenated in English, such as "anti-ferroptosis" and "anti-radiation". In scientific English, these words are usually written without hyphens unless the hyphen is necessary to avoid ambiguity.

2.      When presenting comparisons with other plants, it would be helpful to briefly discuss the implications of these nutritional differences, especially in terms of potential practical applications or health benefits.

3.      Maintain consistency in formatting references, tables and figures. This includes uniform use of citation styles and formatting of table and figure captions.

4.      Contextualizing the results within the broader field of herbal medicine and traditional Chinese medicine may help position O. violaceus as a promising candidate for future pharmaceutical developments.

5.      Future Implications: Discussing the implications of these findings for future research, possible clinical applications, and the development of new medicines can offer valuable insight into the plant's potential impact.

6.      Discussing potential unexplored industrial or commercial applications based on the unique properties of O. violaceus may offer inspiration for future research and development.

7.      Although the article mentions the need for more research into monomeric compounds and their mechanisms of action, specifying areas, where data are particularly sparse or contradictory, can help direct future research efforts.

In general, the manuscript appears to be well structured and informative, covering important aspects of the plant under study. Incorporating the aforementioned suggestions can help further improve the quality of the manuscript.

Author Response

Thanks very much for taking your time to review this manuscript. All your suggestions are of great significance to my thesis writing and scientific research work. We have studied comments carefully and have made corrections which we hope meet with approval. Revised portions are highlighted in the manuscript.

Point 1: The presence of hyphens in words that would not normally be hyphenated in English, such as "anti-ferroptosis" and "anti-radiation". In scientific English, these words are usually written without hyphens unless the hyphen is necessary to avoid ambiguity.

Response 1:

Original line 117,18,59,123205,206,250,251,264,276,289,290,374,393,403,405: " anti-ferroptosis, anti-radiation, anti-tumor, anti-oxidation " has been changed to " antiferroptosis, antiradiation, antitumor, antioxidation ". The revised places have been highlighted.

Point 2:  When presenting comparisons with other plants, it would be helpful to briefly discuss the implications of these nutritional differences, especially in terms of potential practical applications or health benefits.

Response 2:

Because of the high content of vitamin C and carotenoids in O. violaceus, we have added the effects of these nutrients on human  healt: "Some studies have illustrated the importance of vitamin c in early infant development, especially in the brain and cognition[19]. In addition, vitamin c deficiency can have a negative impact on infectious diseases, cancer, diabetes, sepsis and cardiovascular dis-ease, among others[20]. Furthermore, β-carotene has been shown to significantly reduce the risk of oral, laryngeal and breast cancers[21]. O. violaceus has a higher overall nutri-tional value compared to other vegetables in the Brassica genus, so it has great potential application prospects.". The revised places have been highlighted in page 3 line 102-108.

Point 3: Maintain consistency in formatting references, tables and figures. This includes uniform use of citation styles and formatting of table and figure captions.

Response 3:

We have changed the format in the revised draft. Please see in Table 8,Table 9, Table 10, Table 11, Table 12, Table 13 in the revised manuscript.

Point 4: Contextualizing the results within the broader field of herbal medicine and traditional Chinese medicine may help position O. violaceus as a promising candidate for future pharmaceutical developments.

Response 4:

At present, traditional Chinese herbs and chemical drugs are often combined in clinical treatment. According to the antiradiation activity of O. violaceus, the possibility of combining with conventional chemical antiradiation drugs was revealed in the future. A discussion of this issue: " O. violaceus is a kind of medicinal resource plant with high development value because of its rich nutritive value and health care function. It contains quercetin, catechin and other antioxidant active ingredients. The alkaloids of O. violaceus have great benefits to human health, and can be used to treat diseases such as antiinflammatory, antioxidant and antitumor, and have a protective effect on the liver. In particular, the new skeleton alkaloid orychophragine D shows significant antiradiation activity. Herbs are rarely used alone and are often combined with other herbs in Chinese medicine theory. Over recent years, there has been a growing trend of integrating herbs with conventional chemical drugs in clinical practice, resulting in significant therapeutic benefits. Thus O. violaceus may have a superimposed effect when used in conjunction with conventional anti-radiation drugs. " is discussed in page 26, line 761-771 in the revised draft.

Point 5: Future Implications: Discussing the implications of these findings for future research, possible clinical applications, and the development of new medicines can offer valuable insight into the plant's potential impact.

Response 5:

We have added the discussion: " Through the above findings, we can infer the mechanism of antiradiation and antiferroptosis, reveal the internal mechanism of its traditional curative effect by using clinical key indicators and data changes, and provide scientific basis for disease treatment, which will help position O. violaceus as a promising candidate for future pharmaceutical development." in page 26, line 771-775 in the revised manuscript.

Point 6: Discussing potential unexplored industrial or commercial applications based on the unique properties of O. violaceus may offer inspiration for future research and development.

Response 6:

We have added the industrial applications: " O. violaceus is known to be dense in fatty acids, but has very little current application and potentially very high economic value. O. violaceus oil appears to have special lu-bricating properties due to the dihydroxylation of its major fatty acids and the presence of estolides in the oil. O. violaceus has the potential to be an industrial crop with oil properties distinct from those of castor oil as a chemical feedstock [128]. In addition, it is also rich in vitamins, amino acids, trace elements, etc., which can be used as a special dietary supplement to provide elements that cannot be synthesized by the human body." in page 26, line 778-784 in the revised manuscript.

Point 7: Although the article mentions the need for more research into monomeric compounds and their mechanisms of action, specifying areas, where data are particularly sparse or contradictory, can help direct future research efforts.

Response 7:

We put forward more specific requirements for monomeric compounds, and also proposed the preparation of compounds through chemical total synthesis and the establishment of quality standards system, laying the cornerstone for future industrial production of drugs. The revised places have been highlighted. We have added the discussion: " Further isolation and identification of chemical components can lay the foundation for their pharmacological activities and development applications. The extraction, preparation and purification processes should adopt green chemistry methods to minimize the use of organic reagents and mitigate environmental pollution. Highly active compounds can be prepared by chemical total synthesis, laying the foundation for future industrial production of drugs. In addition, most of the identified compounds have not yet been validated in vivo, despite in vitro activity assessments to indicate that they have pharmacological effects. The in vivo pharmacodynamic evaluation and molecular biological mechanism research are essential to promote traditional and modern drug development. "and " The sources of O. violaceus vary slightly in different regions, but there is no uniform quality standard system yet. Therefore, a comprehensive quality standard system should be established to provide a basis for the clinical application and resource development of O. violaceus. There are still numerous promising opportunities and challenges in the research filed related to O. violaceus in future." in page 27, line 788-797, line 801-805 in the revised manuscript.

Once again, thank you for your sincere suggestion. Heartily, I am looking forward to receiving your affirmative reply.

Reviewer 4 Report

Comments and Suggestions for Authors

Paper is very well written presenting comprehensive info about nutritional value, chemical compositions, pharmacological activity, and application value Orychophragmus violaceus.

Some minor remarks:

1. In Table 3. Amino acid content (mg/g) precise, that it was calculated per one g of protein or that was expressed in micrograms (please re-check);

2. Table 4 do not presents aminoacids content;

3. 3.4. should be "Fatty acids";

4. palmitic acid is not unstaturated (L. 140);

5. Re-check L. 151 (I mean " 3 polyunsaturated fatty acids (PUFAs, Cn: 2,3,60.15%)"); 

6. Fig. 1 presents β-D- glucopyranosyl-12-hydroxyjasmonate, but exact configuration of glucose is missed. Please re-check or draw b-D isomer of glucose;

7. What is "proteinit"?

8. Re-calculate clucosinelate content from mM/g into mg/g (Table 6);

9. Correct structure 23 (Catechin) on Fig. 2;

10. Correct typos in "rhaMnoside";

11. 7,8-Dimethylpyrazine (56) has  not correct structure;

12. Remove terminal hydrogens from 57;

13. Use the same convention for all compound, I mean presented or absent terminal -CH3 group;

14. Instead of "Trans-[3-(4’-hydroxyphenyl)-2-propenoic acid]" use p-hudroxycoumaric;

15. Correct typo in "protocatcchuate".  Also compound is not a salt.

16. Instead of "3,4,5-trimethoxybenzoic acid" use "Eudesmic" acid;

17. Draw proper stereoisomerism in phytosterols moiety;

Author Response

Thanks very much for taking your time to review this manuscript. All your suggestions are of great significance to my thesis writing and scientific research work. We have studied comments carefully and have made corrections which we hope meet with approval. Revised portions are highlighted in the manuscript.

Point 1: In Table 3. Amino acid content (mg/g) precise, that it was calculated per one g of protein or that was expressed in micrograms (please re-check);

Response 1:

We have rechecked that the amino acid content was calculated per one g of protein.

Point 2:  Table 4 do not presents amino acids content

Response 2:

We are very sorry for the error with the title of Table 4. Original Table4: "Table 4. Amino acid content (mg/g)." has been changed to "Table 4. Contents of elements in different organs of O. violaceus (μg/g)". The revised places have been highlighted in page 4.

Point 3: 3.4. should be "Fatty acids";

Response 3:

Original page 6: " 3.4. Fatty acid" has been changed to "3.4. Fatty acids ". The revised places have been highlighted. Please see in line 188 of page 5 in the revised manuscript.

Point 4: palmitic acid is not unstaturated (L.140);

Response 4:

We are very sorry for the mistake. We've removed the "palmitic acid".

Point 5: Re-check L. 151 (I mean "3 polyunsaturated fatty acids (PUFAs, Cn: 2,3,60.15%)

Response 5:

We have rechecked " 3 polyunsaturated fatty acids (PUFAs, Cn: 2,3,60.15%)", and found no error. 3 polyunsaturated fatty acids include linoleic acid (C18:2), linolenic acid (C18:3) and cis-11,14- eicosenoic acid (C20:2).

Point 6: Fig. 1 presents β-D- glucopyranosyl-12-hydroxyjasmonate, but exact configuration of glucose is missed. Please re-check or draw b-D isomer of glucose;

Response 6:

We have modified Figure 1 in the revised manuscript.

Point 7: What is "proteinit"?

Response 7:

We are very sorry for the mistake. Original Table6: "Crude proteinit content(%)" has been changed to " Crude protein content(%)". The revised places have been highlighted. Please see in Table6 of page 6 in the revised manuscript.

Point 8: Re-calculate glucosinolate content from mM/g into mg/g (Table 6);

Response 8:

We have re-calculate glucosinolate content from μM/g into μg/g. The revised places have been highlighted. Please see in Table 6 of page 6 in the revised manuscript.

Point 9: Correct structure 23 (Catechin) on Fig. 2;

Response 9:

We have modified the structure 23 (Catechin) in Figure 2 in the revised manuscript.

Point 10: Correct typos in "rhaMnoside";

Response 10:

We are very sorry for the mistake. Original: " rhaMnoside " has been changed to " rhammnoside". The revised places have been highlighted in revised manuscript.

Point 11: 7,8-Dimethylpyrazine (56) has not correct structure;

Response 11:

We are very sorry for the mistake. Original Table 8: " 7,8-Dimethylpyrazine " has been changed to "7,8-Dimethylalloxazine ".

Point 12: Remove terminal hydrogens from 57;

Response 12:

We have modified the structure 57 in Figure 3 in the revised manuscript.

Point 13: Use the same convention for all compound, I mean presented or absent terminal -CH3 group;

Response 13:

We are very sorry for the mistake. We have unified that absent terminal -CH3 group in the revised manuscript.

Point 14: Instead of "Trans-[3-(4’-hydroxyphenyl)-2-propenoic acid]" use p-hudroxycoumaric;

Response 14:

Original Table9: " Trans-[3-(4’-hydroxyphenyl)-2-propenoic acid]" has been changed to " p-hudroxycoumaric ". The revised places have been highlighted. Please see in Table9 of page 11 in the revised manuscript.

Point 15: Correct typo in "protocatcchuate".  Also compound is not a salt.

Response 15:

We are very sorry for the mistake. Original Table10: " protocatcchuate " has been changed to " protocatechuic acid ". The revised places have been highlighted. Please see in Table10 of page 12 in the revised manuscript.

Point 16: Instead of "3,4,5-trimethoxybenzoic acid" use "Eudesmic" acid;

Response 16:

Original Table10: " 3,4,5-trimethoxybenzoic acid " has been changed to " Eudesmic acid ". The revised places have been highlighted. Please see in Table10 of page 12 in the revised manuscript.

Point 17: Draw proper stereoisomerism in phytosterols moiety;

Response 17:

We have modified Figure 7 in the revised manuscript in page 14.

Once again, thank you for your sincere suggestion. Heartily, I am looking forward to receiving your affirmative reply.

Round 2

Reviewer 2 Report

Comments and Suggestions for Authors

The authors have revised the manuscript as a suggestion.

I have agreed the revised manuscript for publication.